# Measurement report: Altitudinal variation of CCN activation across the Indo-Gangetic Plains prior to monsoon onset and during peak monsoon periods: Results from the SWAAMI field campaign

Mohanan R. Manoj[1], Sreedharan K. Satheesh[1, 2], Krishnaswamy K. Moorthy[2], Jamie Trembath[3] and Hugh Coe[4]

[1]DST-Centre of Excellence in Climate Change, Divecha Centre for Climate Change, Indian Institute of Science, Bangalore, India
[2]Centre for Atmospheric and Oceanic Sciences, Indian Institute of Science, Bangalore, India
[3]FAAM Airborne Laboratory, Cranfield MK43 0AL, U.K.
[4]Centre for Atmospheric Science, School of Earth and Environmental Sciences, University of Manchester, Manchester, UK

Correspondence: Mohanan R. Manoj (manojshibika@gmail.com)

**Abstract.** Vertical distributions (altitude profiles) of condensation nuclei (CN) and cloud condensation nuclei (CCN) and their spatial variations across the Indo-Gangetic Plain (IGP) have been investigated based on air-borne measurements carried out during the SWAAMI field campaign (June to July, 2016) capturing the contrasting phases of the Indian monsoon activity in 2016; just prior to its onset and during its active phase. Prior to the monsoon onset, high concentrations of CN and CCN prevailed across the IGP and the profiles revealed frequent occurrence of elevated layers (in the altitude range 1-3 km). Highest concentrations and elevated peaks with high values occurred over the central IGP. The scenario changed dramatically during the active phase of the monsoon, when the CN and CCN concentrations dropped (CN by 20 to 30% and CCN by 6 to 25%) throughout the IGP with more pronounced changes at altitudes higher than 3 km where decreases as high as >80% were observed. These reductions have an east to west decreasing gradient; being most remarkable in the eastern IGP and very weak over the western IGP where the CN concentrations above 3 km increased during the monsoon. The activation ratios (AR) showed contrasting features, increasing with increase in altitude, prior to the onset of monsoon, reversing the trend to decrease with increase in altitude during the active phase of the monsoon. The supersaturation spectrum became flatter during the active phase of the monsoon indicating an increase in the hygroscopicity of aerosols, following the mixing of surface-based emissions with the advected marine airmass.

## 1 Introduction

Spatio-temporal characteristics of aerosols and their interactions with clouds respectively are key parameters determining the direct and indirect climate forcing by aerosols (Twomey, 1974; Albrecht, 1989; Stocker et al., 2013). Uncertainties associated with the spatial distribution and temporal variations in the physical and chemical properties of aerosols limit our ability to

accurately quantify the climate impact of aerosols. Extensive research in the recent decades have led to significant reduction in the uncertainties by improving the characterisation of aerosols, especially in the perspective of interaction with radiation (Stocker et al., 2013; Bellouin et al., 2020). Nevertheless, the indirect effect of aerosols on climate through aerosol-cloud interactions remains largely uncertain (Stocker et al., 2013; Rosenfeld et al., 2014; Fan et al., 2016). It is fairly well-established

that the radiative effects of aerosols on clouds mostly act to suppress precipitation, through a decrease in the solar radiation reaching the surface (Trenberth et al., 2009). Aerosols reduce the heat available for evaporating water and energizing convective rain clouds, by scattering and absorbing radiation and by modifying cloud properties. Intrusion of a large number of fine aerosols into clouds can inhibit precipitation by slowing down the conversion of cloud drops into raindrops, which might prevent very shallow and short-lived clouds from precipitating (Rosenfeld et al., 2008; Rosenfeld et al., 2014). The

opposite scenario of aerosol-cloud interactions leading to the invigoration of clouds and vigorous precipitation has also been reported, even over moderately polluted environments (Lebo and Seinfeld, 2011; Altaratz et al., 2014; Koren et al., 2014). Thus, activation of aerosol particles for inducing cloud formation remains a topic of intense investigation as aerosols, through their effects on clouds, can induce large changes in precipitation patterns. Changes in precipitation patterns, in turn, would affect the regional water resources as well as the regional and global circulation systems that constitute the Earth's climate

(Song et al., 2014). As such, this knowledge assumes a lot of interest and importance. The fraction of the aerosols or condensation nuclei (CN) acting as nucleation sites for cloud droplets are known as cloud condensation nuclei (CCN). The variations in the CCN properties are strongly influenced by the number size distribution and chemical composition of aerosols (Dusek et al., 2006; Hudson, 2007; Gunthe et al., 2009; Rose et al., 2011). CCN activation depends on the critical diameter required for activation, which in turn depends on the hygroscopicity of aerosols, determined by their chemical composition.

Despite concerted efforts to understand the aerosol-cloud interactions and the associated feedback mechanisms in the atmosphere, large uncertainties still exist (McFiggans et al., 2006; Andreae and Rosenfeld, 2008; Stevens and Feingold, 2009). This mainly arises from the region-specific and heterogeneous nature of aerosols, their vertical mixing and advection to long distances in the real atmosphere, and sparseness of in-situ measurements of the vital parameters of CCN, such as the vertical distribution of the CCN number concentration, CCN efficiency and its variation with supersaturation (SS) (Seinfeld et al.,

55    2016).

Over the Indian sub-continent, the columnar aerosol loading is increasing steadily (Babu et al., 2013) while changes are also observed in the rainfall pattern, with significant decreasing trends in moderate rainfall events and increasing trends in extreme rainfall events (Goswami et al., 2006; Guhathakurta et al., 2015). Since aerosols modulate the monsoon circulation and rainfall distribution (Gautam et al., 2009), the observed changes in aerosol loading and precipitation patterns are of utmost interest.

Studies from different parts of India, tried to understand the properties of the regional aerosols like their chemistry, CCN activity and hygroscopic properties. However, most of these studies were ground-based focusing on case studies or long-term measurements on seasonality. Studies from Kanpur, in the central IGP, looked into inter-seasonal variability due to changes in the size and chemical composition of aerosols (Patidar et al., 2012; Bhattu and Tripathi, 2014), and hygroscopic nature of aerosols (Bhattu et al., 2016) while Ram et al., (2014) showed the role of primary and non-hygroscopic aerosols in the

weakening the diurnal variations of the CCN activation despite large variations in the CN and CCN concentrations. Arub et al. (2020) characterized the chemical composition and size distributions of aerosols in Delhi, situated in the central IGP, and showed the impacts of air masses coming from different locations on the hygroscopicity and CCN activation. The seasonal variations of CCN properties revealed significant influence of aerosols transported from the IGP in modulating the aerosol cloud interactions over the Nainital, located in the Himalayas (Dumka et al., 2015; Gogoi et al., 2015). Similar observations were also reported from Darjeeling, a high altitude site in the eastern Himalayas (Roy et al., 2017), revealing the impact of aerosol transport from the IGP on the CCN properties. The observations from a high altitude site, Mahabaleshwar, located in the southern part of India, revealed large seasonal variations of CCN and significant role of organics in the CCN activation (Leena et al., 2016, Singla et al., 2017). The observations from Gadanki, in south India (Shika et al., 2020), revealed a shift in the regimes of CCN activation and cloud formation, from pre-monsoon to monsoon season which has important implications on cloud droplet formation. Measurements of CN and CCN over a rain shadow region Solapur, in central India carried out as part of the CAIPEEX campaign (Jayachandran et al., 2020a), revealed variations in the CCN characteristics within the monsoon period. Jayachandran et al (2017) characterised the CCN properties at Thiruvananthapuram, a coastal location in south India, while Jayachandran et al (2018) reported contrasting characteristics of CCN properties over Ponmudi, a hill station in Thiruvananthapuram in comparison to a nearby coastal location.

In this context, improved knowledge of the vertical distribution of CCN properties is all the more important. However, such measurements are not abundant globally, and are sparse over the Indian region; though a few air-borne direct and indirect measurements have been made in the recent years. Using air-borne lidar measurements, Satheesh et al. (2008) have shown the role of elevated aerosol layers over the Indian peninsula in producing large heating at 4 to 5 km altitude, just above the low-level clouds during pre-monsoon season. During 2009, airborne atmospheric measurements of vertical profiles of aerosols, clouds and meteorological parameters extending up to 7 km above sea level (asl) were carried out over different parts of India as part of Cloud Aerosol Interactions and Precipitation Enhancement EXperiment (CAIPEEX) (Prabha et al., 2011; Kulkarni et al., 2012; Padmakumari et al., 2013). In-situ measurements of CN and CCN were also carried out during the Indian Tropical Convergence Zone (ITCZ) campaign, during the monsoon period in 2009 (Srivastava et al., 2013). However, the majority of these measurements remained confined to different phases of the monsoon season in different years so that the changes in the CCN characteristics as the season transits from dry summer to wet monsoon season remained largely elusive.

The joint Indo-UK field experiment, SWAAMI was formulated in the above backdrop, aiming at filling the knowledge gaps related to the properties of aerosols over the Indian region and its relation to the Indian Summer Monsoon (ISM) by making in situ measurements of aerosol and cloud properties, immediately before monsoon onset and during its active phase using instrumented aircraft (Manoj et al. (2019) and references there-in). The campaign was executed jointly by the Ministry of Earth Sciences (MoES) of India, the Indian Space Research Organisation (ISRO) and the Natural Environment Research Council (NERC) of the UK and employed UK's BAe-146-301 Atmospheric Research Aircraft. This was closely preceded by airborne measurements of aerosol properties across the IGP aboard an Indian aircraft up to an altitude of ~3.5 km (Jayachandran et al 2020a; Vaishya et al., 2018). Our study, provides altitude resolved CCN properties at higher altitudes compared to

Jayachandran et al, (2020a) and explores the impact of chemical composition reported by concurrent measurements (Books et al., 2019a) on the CCN properties. The campaign details along with the measurement protocols are briefly stated below (more details are available in Manoj et al. (2019)) followed by the results and discussions.

## 2 Data and methodology

The airborne measurements of the SWAAMI campaign was carried out in three phases extending from 11-June-2016 to 11-July-2016 (Brooks et al., 2019a; Manoj et al., 2019), aboard the BAe-146-301 atmospheric research aircraft operated by the Facility for Airborne Atmospheric Measurements (FAAM) BAe-146 (Highwood et al., 2012; Johnson et al., 2012). During these, data were collected over an altitude range from very close to the surface to as high as 8 km. This study reports the measurements carried out during phase 1 (just prior to the onset of the Indian summer monsoon (ISM), hereinafter called 'pre-onset') and phase 3 (during its active phase) across the IGP (Indo-Gangetic Plains) covering its western, central and the eastern parts and capturing the transition of aerosol properties from dry to wet climatic zones. The measurements also covered parts of central India. Details of the measurements are summarized in Table 1. The airborne measurements focussed on three locations, Lucknow (LCK - 26.84 °N, 80.94 °E, 126 m asl) in the central IGP, Jaipur/Jodhpur (JPR - 26.91 °N, 75.78 °E, 431 m asl; JDR- 26.23 °N, 73.02°E, 233 m asl) located at the southern tip of the western IGP and Bhubaneswar (BBR – 20.29 °N, 85.82 °E, 58 m asl) located just south of the eastern boundary of the IGP along the eastern coast of India; though the base station for all the measurements was Lucknow. Measurements were also made around two locations in central India, Nagpur (NGP - 21.14 °N, 79.08 °E, 310 m asl) during Phase 1 and Ahmedabad (AMD - 23.02 °N, 75.57 °E, 53 m asl)) during Phase 3.

The phase 1 measurements covered regions around Lucknow making east-west transects across the IGP covering Jaipur (west) and Bhubaneswar (east) and proceeded to the central Indian region, Nagpur. The third phase covered the above transect (except Nagpur), and made additional measurements near Ahmedabad. During measurements the aircraft maintained a typical ascend rate of 5.5 m s$^{-1}$ and descend rate of 6.5 m s$^{-1}$, thereby providing data at a high vertical resolution of ~7 m. The horizontal velocity of the aircraft has been typically ~100 m s$^{-1}$, which means that a typical ascent of the aircraft from near the surface to ~7 km covers a horizontal distance of roughly 130 km. The descriptive statistics have been performed after separating the data vertically into 300 m blocks. During the campaign the aircraft made 22 dedicated scientific flights spanning approximately 100 hours in three phases: Phase-1: 11-Jun-2016 to 13-Jun-2016 (3 flights; ~12 hours; just prior to onset of monsoon); and Phase-3: 02-Jul-2016 to 11-Jul-2016 (9 flights: ~40 hours, during active phase of the monsoon), while Phase 2 was meant for other objectives, not related to aerosols. A list of the aerosol instruments aboard, the parameters retrieved from the measurements, the relevant reference to the principle of instrument and data deduction details, general aircraft data and met data is provided in Manoj et al. (2019) and references therein. The detailed flight tracks are given in the supplementary material (figures S1 and S2).

The aerosols sampled in this study were collected using a Rosemount inlet. The Condensation Nuclei (CN) concentration was estimated using a modified water filled Condensation Particle Counter (CPC) TSI 3786. Operating at a flow rate of 0.6 Litres Per Minute (LPM), it can detect particles in the size range 2.5 nm to >3 μm and can measure concentrations up to $10^5$ particles cm$^{-3}$. The Cloud Condensation Nuclei (CCN) concentration was measured using a dual column cloud condensation nuclei counter (Droplet Measurement Technologies Inc.  CCN-200), which is a continuous-flow stream-wise thermal gradient

chamber (CFSTGC) instrument. The CCN counter operated at a flow rate of 1 LPM and the flow is evenly split between the two columns. The sample to sheath flow ratio is set to 1:10 which leaves 0.05 LPM for each column for sampling. The samples flowing to the CCN counter and CPC, first passes through a Nafion dryer (Permapure MD-110-12S) which prevents condensation in the sample lines. The mean RH of the ambient and sample lines during pre-onset were 49.7% and 28.7 % and those during monsoon measurements were 73.5% and 54.4% respectively for ambient and sample lines. To gain as much

information as possible from a flight the instrument was set up to have one column scanning three different supersaturations (0.12%, 0.23% and 0.34%) and the other was stable (0.1%). providing four different supersaturations every 15 mins. More details are provided in the supplementary material. Details of the principle of operation of the CCN counter are available elsewhere (Roberts and Nenes, 2005; Lance et al., 2006). More details on the inlet, the plumbing and working of the CPC and CCN counter are given in (Trembath, 2013).

**3 Synoptic meteorology during the campaign**

In 2016, the monsoon onset was delayed by a week; the onset at the southern tip of the Indian peninsula occurred on 8th June. The rainfall for the season however was near normal; about 97% of the long-term average for the season (Purohit and Kaur, 2016). Consequent to this delayed onset, monsoon arrived at LCK only on 21st June and covered the entire Indian region by 13th July (climatologically, this should occur towards the end of June). The meteorological wind fields at 850 hPa from the

ERA-Interim reanalysis of the European Centre for Medium Range Weather Forecasts (Dee et al., 2011) was used  to show the advance of the southwest monsoon during the two phases of measurements (Figure 1).

The spatial distributions of accumulated rainfall for the respective phases, as well as for the preceding one-week are shown in Figure 2, based on the high-resolution (0.25°x0.25°) rain data (Pai et al., 2014) provided by the Indian Meteorological Department during the campaign period.  The total rainfall (in mm) received over the northern part of India (>25°N) during 4-

10 June (one week before the commencement of phase 1 of the SWAAMI campaign, Figure 2a) resulted from a few isolated rain events to the north of JPR and LCK leaving Central India and majority of the IGP without any rain. Some isolated rain events occurred in the vicinity of NGP, while BBR received moderate rainfall prior to the phase 1 measurements. The accumulated rainfall during phase 1 measurements 11-13 June) is shown in Figure 2b. Apart from the isolated rainfall in the vicinity of JPR and NGP the other locations hardly received any rain.

However, by the third phase of the campaign, the monsoon was established in the northern parts of India and IGP. The total rainfall received in the vicinity of the measurement regions during the period 25$^{th}$ June to 1$^{st}$ July i.e., 1 week before the phase 3 measurements are shown in figure 2c. JDR and BBR received moderate rainfall during this period while LCK, JPR and

AMD received very less rain. But during the phase 3 measurements the main regions under study received weak/ moderate rainfall (figure 2d).

**4 Results and discussions**

**4.1 Observations during Phase 1 – Pre-onset Phase**

As stated earlier, during this phase, the measurements were made centred about LCK, JPR, BBR and NGP. Consequent to the delayed onset, monsoon did not advance even to the southernmost of these locations (NGP) prior to phase 1 (Figure 1a). As such, phase 1 measurements corresponded to conditions just prior to onset of the monsoon (pre-onset) at all the locations and

across the IGP. The mean vertical profiles of CN and CCN concentrations, were estimated using 5 profiles for LCK and 2 profiles each for the other three locations. The altitude variations of the mean concentrations of CN and size-integrated CCN at 0.1% SS and the corresponding activation ratio (AR) at 0.1% SS, for all the 4 locations are shown in figure 3. The mean supersaturation was 0.099 ±0.005, with 95% of the points within the interval 0.098 and 0.100. The activation curves for ambient aerosols will not follow that of ammonium sulphate particles if they are externally mixed and for SS=0.1% the dry diameter

of ambient aerosols would be in the range 75 – 125 nm (Deng et al., 2011). The CCN number-size distribution curve will peak above 200 nm with the maximum size of the CCN close to 600 nm at SS=0.1% (Gunthe et al., 2009).

Being the centre of operations, measurements were made around LCK (representing the Central IGP) on all the three days (11, 12 and 13 June). As such, the mean profiles of CN, CCN and AR, shown in Figure 3a provides the best statistical dataset (spatially and temporally averaged) for pre-onset conditions over the Central IGP. The highest CN concentration (>3500 cm$^{-3}$

$^{3}$) occurred near the surface and decreased rather monotonically towards higher altitudes to reach values of ~1800 cm$^{-3}$ and ~ 100 cm$^{-3}$ respectively at 1 km and 6 km. Variations (represented by the standard deviations) are higher closer to the surface. Vertical variation of CCN concentration (blue line) followed the pattern observed in CN up to an altitude of 1.5 km, and above that CCN showed an increase with a couple of peaks (marked by the ellipse in the figure) in the altitude range 1.5 to 3.5 km. At higher levels, the variation of CCN again followed the pattern of CN. The activation ratio was low (~0.08) near the surface

and remained steady until about 2 km, above which it increased rather sharply to reach value of ~ 0.15 at 3 km and remained nearly steady towards higher altitudes. This sudden increase in the AR appears to be responsible for the observed peaks in CCN concentration (despite the decrease in CN) in the altitude range of 1.5 km to ~2.5 km and is indicative of the presence of a different aerosol type prevailing above ~1.5 to 2 km, which is more CCN active in nature. CCN concentrations are more sensitive to hygroscopicity in comparison to mixing state at low SS (~0.15%) (Meng et al., 2014).

The nearly concurrent characteristics of CCN over the semi-arid region of the western IGP are examined in Figure 3b based on the measurements around JPR on 11$^{th}$ June. While the nature of the altitude variation of CN and CCN is similar to that observed at LCK, the overall CN and CCN concentrations are lower, the elevated peaks (occurring above 2 km) are sharper

and seen on both CN and CCN concentrations. Again, the activation ratio is slightly higher (~ 0.1 closer to surface) and increases steadily with altitude, reaching around 0.15 at 3 km. However, here the data is limited up to about 3.2 km only.

Moving over to the eastern IGP, represented by BBR, Figure 3c shows the altitude variations based on measurements made on 12[th] June. Differing slightly from the pattern seen at the western and central IGP regions, the vertical variations in AR are weaker in this region. In general, CN and CCN showed variations similar to those at LCK and JPR with higher values near the surface and lower values at higher altitudes. The highest CN/CCN concentration (2264/184 cm$^{-3}$) was observed near the peak below to 1 km. Two elevated peaks (the first near 1 km and the second peak near 3 km) occur in the concentrations of both CN and CCN. The second peak is much broader, extending from 2.5 km to 3.5 km, with high CN and CCN concentrations of 1845 and 161 cm$^{-3}$ respectively. The activation ratio at lower altitudes (~0.1) is higher than the corresponding values at LCK, and closer to those seen at JPR. However, there is little variation in the AR up to an altitude of 4.5 km.

Altitude variations of the mean CN concentration, CCN concentration and AR over the Central Indian region, from measurements around NGP, made on 13[th] June between 12:45 to 13:45 local time, are shown in figure 3d. The profiles show a nearly steady CN concentration in the lower altitude region of 500 to 2500 m, and decrease thereafter reaching low values close to 200 cm$^{-3}$ at ~4.2 km, and then remaining nearly steady up to 5.7 km. In the lower altitude region, CCN decreases with altitude, and then reveals a peak around 2.5 km and then decreases monotonically. The activation ratio remained low (~0.08) and nearly steady up to 2.5 km and then increased to 0.15 above 3 km (somewhat similar to the observation over IGP).

### 4.1.1 Regional features of CCN during pre-onset phase

The regional picture of the altitude variations of the CN, CCN (SS=0.1%) and AR (SS=0.1%) for the pre-onset phase is shown in figure 4. The most striking feature is that there is a steady increase in the AR with altitude in the western and central IGP, the altitude variations are much weaker in central India and almost absent in the eastern IGP indicating a difference in the aerosol properties across the IGP. Below 2 km the CN concentrations are highest in the central IGP (where the anthropogenic emissions are higher), closely followed by eastern IGP (BBR) and western IGP (JPR), with the lowest values in central India (NGP). The CN concentrations above 2 km, are comparable at all the stations except BBR, where the CN concentrations were high due to the presence of a thick elevated layer in the 2.5 to 3.5 km region. The main factor controlling the CCN activation at low SS (~0.1%) is the hygroscopicity of the particles (Meng et al., 2014), which is determined by its chemical composition. Gunthe et al. (2009) found that particles having diameter ~50 nm and ~200 nm respectively had hygroscopicity parameters 0.1 and 0.2 respectively.

The altitude variations of CCN concentrations are not as well defined as those of CN. Below 1.5 km the CCN concentrations were highest in eastern IGP (except for a sharp drop in the concentrations near 0.5 km) followed by LCK, JPR and NGP. CCN concentrations increase in the 1.5 to 2.5 km range in all data except those from NGP. Above 2.5 km, the magnitudes of CCN at various locations differ widely (though with an overall decreasing trend with altitude), with highest concentrations at BBR followed by LCK. The lowest concentrations were observed at NGP where the values were comparable to JPR in the 3 to 3.2 km altitude range; measurements above 3.2 km are not available for JPR.

The AR (at SS=0.1%) clearly indicates two features: (a) the presence of more CCN active aerosols at higher altitudes at western and central IGP as well as central peninsula, which is not seen over eastern IGP (b) over the most anthropogenically impacted central IGP, two different aerosol types with a less CCN active layer below ~2 km, and a more CCN active layer above leading to a change in the CCN/CN ratio in the altitude region 2 to 3 km. The vertical distribution of the concentrations and the AR of aerosols had more distinct regionally varying patterns above 3 km. The increase in the activation ratio above 3 km observed in our study has not been reported earlier. The regional variations in the CCN concentrations above 3 km were found to have an east to west gradient, with highest values in the east. To understand the peculiar behaviour of AR over the eastern IGP with two elevated peaks in the CN and CCN (near 1 km and 3 km), and a nearly steady and low value of AR, we examined the airmass back trajectories in Figure 5. These clearly reveal that the altitude region of 1.5 to 3 km was under the influence of distinct advection pathways. One path favours the advection of more hygroscopic particles (as evidenced by figs 3 a and b) from the northwest region (magenta colour in Figure 5), at around 3 km altitude, while the other encounters less CCN active aerosols being lofted from close to the ground, from sources in the southwest (central IGP), as it approached BBR (red line in figure 5). The presence of less hygroscopic particles (probably freshly emitted BC) in the lower altitudes <1 km is seen from Figs 3a & 3d (around central IGP and central peninsula). The mixing of these two types of particles in the altitude region of 1.5 to 2.5 km seems to contribute to the elevated peaks in CN and CCN as well as the near steady value of AR in this region, in contrast to the increase seen at other locations. From an independent airborne measurement of CCN characteristics (again as a part of SWAAMI, using an Indian aircraft) during the first week of June, a week prior to our measurements, Jayachandran et al. (2020a) have observed almost similar features in the Eastern IGP; with the CCN concentrations (at 0.4% supersaturation) below 1 km being comparable to that at central IGP, while in the 1 to 3 km range the CCN concentrations were higher compared to central IGP. The complex interplay of different advected species appear to be responsible for this, at least partly. The results by Jayachandran et al. (2020a) are however limited to the changes happening within the boundary layer and fails to completely capture the elevated aerosol layers which we found to exist even above 3 km.

It is well known that the CCN concentration strongly depends on the number size distribution and chemical composition of aerosols, while the supersaturation spectra depend on the aerosol number size distribution (Fitzgerald, 1973). If ammonium sulphate and adipic acid with size of 100 nm are considered the former can be activated at 0.15% SS while the latter can only be activated at 0.27% SS (Hings et al., 2008, Zhang et al., 2012). As such, we examined the changes in the chemical composition of aerosols at BBR, from concurrent measurements by other investigators aboard the same flight (Brooks et al., 2019a), who have reported a sharp increase in the concentration of organics near 1 km (Brooks et al., 2019a; Figure 11c; B957 PM), where our observations show a sharp increase in CCN concentration and a weak increase in AR, which otherwise remained featureless (nearly steady at ~ 0.08) in the entire altitude region. Moving to higher altitudes, measurements by Brooks et al. (2019a, same figure) have shown a large increase in the concentration of sulphates, NH4, organics and BC in the altitude region of 2.5 to 3 km; where our observations show a broad peak in CN and CCN concentration, but with no perceptible impact on AR. Residential sectors are the main source of organics in the region while brick kilns (which use coal and lignite) emit large amounts of sulphates (Pandey et al., 2014). We hypothesize that though the increase in the concentrations of sulphates

and organics in this region was favourable for an increase in AR, the simultaneous increase in BC (hydrophobic) prevented any conspicuous impact on AR. This is also supported by the observations by Jayachandran et al. (2020a) of low CCN efficiency associated with high concentrations of BC based on independent measurements made a week prior to our measurements. While Jayachandran et al. (2020a), showed that the presence of BC can reduce the activation efficiency of aerosols in the boundary layer we found that this also happens at higher altitudes. Consequently, though CN and CCN increase

in-line with the increase in the concentration of precursor gases (Brooks et al., 2019a), AR remains nearly unaffected. This did not happen for the peak around 1 km, because of the lower concentration of BC at that altitude (Brooks et al., 2019a). Extending the above role of chemistry to the central and western IGP regions, we recall that Brooks et al. (2019a) have shown that within the boundary layer, the concentration of organics (43%) exceeded the concentration of sulphates (29%) in the submicron mass over the central IGP, whereas in the western (JPR/JDR) and eastern (BBR) IGP, sulphate was the dominant species contributing

44% followed by organics (30%). However, this distinction was confined within the boundary layer, above which sulphates dominated throughout the IGP region. This implies that in the central IGP, local emissions contributed significantly at the lower altitudes. This, at least partly, accounts for the increase in the AR above the boundary layer seen both at western and central IGP seen in figs 3a & 3b and fig 4 (c). The low activation within the boundary layer, in the central IGP, appears to be at least partly associated with the high concentrations of BC (which is hydrophobic in nature when freshly emitted), emitted

by the local sources. Similar observations are also reported by Jayachandran et al. (2020a), from near-concurrent measurements at Varanasi in the central IGP (close to LCK) aboard another aircraft about one week prior to our measurements. Thus, we see that along with advection, the changes in the chemistry (concentration of precursors) also has a significant role in producing the observed spatial variation of the altitude profile of CCN characteristics and AR across the IGP prior to onset of monsoon. During the pre-monsoon period, the entire IGP is extremely hot with temperatures routinely above 40°C, going as high 48°C

during the peak. The resulting strong convective mixing distributes local surface-based emissions deep into the Planetary Boundary Layer (PBL), which itself is deep (going to 2 km or more). Above the PBL, however, long-range transport has a strong influence on the altitudinal distribution. We examined the 3-day back-trajectories of the air mass reaching LCK (Central IGP) and JPR (western IGP) in Figures S3 to S6 (supplementary materials). The figures clearly show that while long-range transport had negligible role below 1 km, it influenced significantly above 2 km. This lends further support to our inference

that in the Central IGP the local emission of fresh hydrophobic particles (like BC) is responsible for the low AR within the PBL (<2 km), the long-range transported aerosols including dust, discussed in more detail below, leads to the increase in AR and a consequent increase in CCN (despite the decrease in CN) above 2 km. Jayachandran et al. (2020a) does not cover the transport of aerosols above the boundary layer, which we found were more hygroscopic and more amenable to CCN activation compared to the boundary layer aerosols.

Examining the size distributions, it is well-known that during the pre-monsoon, accumulation mode and coarse mode particles dominate the aerosol volume size distribution respectively within and above the PBL in the Central IGP, due to the influence of the prevailing wind, flowing from the arid, dusty regions in the NW of India (Gautam et al., 2011). The accumulation mode number concentration was greatest within the boundary layer, coinciding with high organic aerosol loading. However, in the

western IGP, coarse mode (dust) aerosols (emitted locally) prevailed at all heights. Concurrent measurements by Brooks et al. (2019b) have shown that the BC within the boundary layer was not coated thickly (rather freshly emitted) but above the boundary layer the BC had thick coating (aged BC) both at Central and western IGP (LCK and JPR). The relatively larger particles above the boundary layer are better amenable for activation compared to the smaller particles within the boundary layer. Observations from the central Himalayan site, Nainital, located 1958 m above sea level (Dumka et al., 2015; Dumka et al., 2021) revealed that aged and coated BC aerosols (consequently bigger in size), transported to this location from the IGP, were more hygroscopic compared to freshly emitted BC aerosols. In the western IGP the CN and CCN variations go hand in hand with a high correlation coefficient of 0.94. The AR is high, and comparable to the values in the central IGP. The increase in AR with altitude is also observed in the dust dominated western IGP, similar to central IGP. Though freshly emitted dust is hydrophobic, they become CCN active when mixed with other species like sulphates and nitrates (Kelly et al., 2007). It may also be noted that the air mass arriving at this region has considerable overpass above the Arabian Sea gathering moisture and allowing the aerosols to mix with marine aerosols, thus enhancing their hygroscopicity (Fig S3). The relation of the observed changes in AR with the chemical composition, BC mixing state and size distribution changes are further investigated in the coming sections also for the observations during monsoon. The low values of CN and CCN near NGP (central peninsula), might have been partly due to the rain in the vicinity, prior to our measurements. The AR has a weak altitudinal variation with values increasing above 3 km indicating a different type of aerosols, but other measurements are not available to further investigate the observed changes.

To summarise the pre-onset scenario:

- Elevated aerosol layers are present throughout the IGP region, above 2 km, and are identified by large increases in the number concentrations of CN and CCN. However, the AR in these elevated layers did not show sharp changes, implying that the sources are more or less homogeneous, at all altitudes within a region, except perhaps at the central IGP.

- The eastern and central IGP had higher concentrations of CN compared to western IGP. Below 1 km the concentrations of aerosols were higher in the central IGP, compared to eastern IGP, while above 2 km, there was a reversal in the pattern. This is associated with the transport of aerosols over BBR, which includes dust from west Asia and anthropogenic emissions from the central IGP. The high wind speeds in the region, during this period, are ideal for transport of pollutants.

- The altitude distribution of CCN, somewhat differed from that of CN. In the altitude range 1.5 to 2.5 km sharp changes are observed in the CCN concentrations throughout the IGP. But above 3 km an overall decreasing trend is observed, with distinct regional variations giving rise to a large westward (decreasing) gradient across the IGP. The complex interplay of local emissions and advection of aerosols, along with the PBL dynamics and chemistry involving precursor gases are found to be responsible for the observed spatial variation of the altitude profiles of CCN characteristics across the IGP, as discussed above.

- The AR values along the IGP and central peninsula are comparable below 2 km and the altitude variations follow a similar pattern. However, above 2 km, two distinct patterns are observed: 1) AR increasing with altitude in the central and western IGP and 2) AR remaining almost steady in the eastern IGP. This shows the presence of more hygroscopic aerosols at higher altitudes in the western and central IGP as well as the central peninsula, but not in the eastern IGP.

- In the eastern IGP, concurrent variations of AR and chemical composition, revealed that the presence of organics and sulphates favoured the activation of CCN at lower altitudes. It was also observed that, despite having high concentrations of organics and sulphates at higher altitudes, the activation of CCN was reduced by an increase in the concentration of (hydrophobic) BC.

### 4.1.2 CCN Characteristics during the Active phase of the monsoon

The above characteristics of CCN are re-examined during the active phase of monsoon, based on the FAAM data collected during Phase 3 of the campaign; covering the different sub-regions of the IGP (east, central and west – BBR, LCK, JPR) and AMD during $2^{nd}$ -$7^{th}$ July. By this date the monsoon is established over the entire Indian region (Figures 1 and 2). The mean profiles of CN and CCN concentrations were estimated using 14 profiles for LCK, 11 profiles for JPR and 2 profiles each for the other two stations as detailed in Table 1. The variation of the mean CN concentration, CCN concentration and the activation ratio (AR) for all the above 4 locations are presented in figures 6 (a-d) following the same methodology as in Figure 3.

The changes from prior to onset of monsoon to active phase of monsoon are clearly depicted in Figure 7 where all the three parameters are compared. Most conspicuous features revealed by Figures 6 and 7 are:

1. Very large reduction in the concentrations of CN and CCN during the active phase (from the values which prevailed prior to the onset) at all altitudes and across the entire IGP; with concentrations of CCN dropping more dramatically than the CN.

2. The effects are most prominent over the eastern IGP followed by the central IGP, while over western IGP, it is rather weak and is significant only at higher altitudes (Fig 7).

3. Reduction in concentrations during the active monsoon phase increases with increasing altitude; from nearly 30% reduction near the surface to as much as 90% at around 4 km altitude (from the corresponding values prior to the onset of monsoon); with the spatial features described in point #2 above.

4. In the semi-arid western IGP, which experiences much lesser rainfall, the reduction (from the values prior to the onset of monsoon) in CN and CCN are marginal and is seen only below ~ 3 km. The average accumulated rainfall over a week prior to the measurements in a 2°x2° grid surrounding JPR, was merely 28 mm. At higher altitudes, the concentrations are comparable to or even higher than those that existed prior to the onset of monsoon, indicating the strong prevalence of long-range transported dust over that region.

5. Examining Figure 7c, it is clearly seen that, in the lower altitudes (below 2 km) the activation ratios are, in general, higher than their corresponding values prior to onset of monsoon across the entire IGP. Above 2 km, there is a reversal in the activation ratio pattern, whereby the increasing trend prior to monsoon being replaced by a decreasing trend

during the active phase. There are, of course, sub-regional differences; with nearly steady values in the Central IGP up to about 4 km, whereas AR decreases conspicuously in the eastern and western IGP regions. The very high values of AR close to the surface at BBR is due to the advection of marine aerosols by the monsoon winds.

During phase 3, there was an additional profiling on 7[th] July in the western India, over AMD, south off the western IGP. The mean features, shown in figure 6d, are (a) a decrease in CN concentrations up to about 2km and nearly steady profile above up to about 5 km and (b) a monotonic decrease in CCN concentration and AR from close to surface to the highest altitude (5 km); somewhat resembling the pattern seen at the western IGP, but with stronger altitudinal variation of AR.

While the CN concentrations are almost the same in the western stations (JPR and AMD) at all altitudes, the CCN concentrations are higher in AMD (figs 6 (b & d)) indicating a more CCN active aerosols. This is attributed to the stronger advection of marine aerosols from the Arabian Sea, bringing in more hygroscopic aerosols to AMD (Fig S7).

### 4.1.3 CCN spectra

With a view to furthering the above understanding of the spatial and vertical variation in CCN characteristics and the changes with respect to the monsoon activity, we have examined the CCN spectra (variation of CCN concentration as a function of supersaturation) using the measured data. It is well-established that the ability of an aerosol to be a CCN is a function of both the hygroscopicity and size distribution of aerosols (Twomey and Wojciechowski, 1969; Hegg et al., 1991; Khain, 2009; Jefferson, 2010). Following Twomey, (1959) and Cohard et al., (1998), we have parameterised the CCN spectra using a power law relation.

$$N_{CCN} = C \, SS^k \tag{1}$$

where $N_{CCN}$ is the number concentration of CCN at a particular supersaturation (SS) and C and k are empirical coefficients. Lower values of k imply quick activation of CN even at low SS; and are generally associated with more hygroscopic and coarse mode aerosols (such as sea-spray). On the other hand, higher values of k mean more activation only at higher SS, typical to less hygroscopic and fine mode anthropogenic aerosols (Hegg et al., 1991; Jefferson, 2010). The altitude variation of k will have implications for aerosol cloud interaction through hygroscopicity and size distribution of aerosols (for example, Raga and Jonas (1995)).

The CCN spectra are shown in Figure 8, where the panels from left to right show the spectra across the IGP from west to east; in each case the top panel represents the conditions just prior to the onset of the monsoon (Phase 1 of the campaign) and the bottom panel represents the active monsoon (phase 3) conditions. The spectra are limited only up to SS = 0.4, as the SS range during measurements were restricted in the range 0.1% to 0.4%, which makes it possible to make high resolution measurements even at higher altitudes. The lines are regression fits to equation (1). The figure reveals the following:

1. In general, irrespective of the phase of the monsoon, k values are the highest in the Central IGP (LCK); though the values during the active phase of the monsoon are lower than the values just prior to the onset of the monsoon. This confirms the prevalence of submicron aerosols with lower hygroscopicity over the central IGP and is also in line with

the high concentration of anthropogenic aerosols in that region (denser sources of emissions). This is also supported by the high values of BC (~2 μg m$^{-3}$) that prevailed over this region as has been reported by Brooks et al. (2019a), from concurrent measurements and long term observations from Nainital (Dumka et al., 2021). As the CCN concentrations at higher supersaturations are mainly attributed to accumulation and fine-mode particles (Lance et al., 2013), the higher values of k over the central IGP clearly suggests prevalence of an aerosol system dominated by fine-mode particles during both phases of the monsoon.

2. The significant reduction in k value over the central IGP during the active phase of the monsoon from its value prior to onset of monsoon is indicative of a change in the aerosol composition brought about by wet removal (including BC, the concentration of which dropped to half its value during the pre-onset phase; Brooks et al., 2019b). The advected moist marine airmass (from the Bay of Bengal and Arabian Sea by the favourable monsoon winds) also contributes to the reduction in k and increase in hygroscopicity (Pringle et al., 2010), in the lower altitude regions. However, the less hygroscopic aerosols, advected by the continental airmass, prevailed at the higher altitudes (>3 km), leading to a decrease in the activation efficiency and increase in k values, even during the active phase of the monsoon.

3. Over the eastern and western IGP regions where, in general, more coarser particles exist (mineral dust over western IGP and marine aerosols over eastern IGP), k values are in general lower than those seen in the central IGP during both the phases of the monsoon activity implying that the aerosols over these regions are amenable for easier activation to CCN compared to those over the Central IGP.

4. However, the responses of k to the distinct phases of monsoon activity provide a contrasting picture over the western and eastern IGP regions. While there is a dramatic reduction in k (from 1.25 to 0.43) in the eastern region, brought in by in increased abundance of marine aerosols here (advection from the Bay of Bengal, due east off BBR), over the semi-arid regions of the western IGP, k has increased (though weakly) to 0.93 from its value (0.81) during pre-onset phase. A change in the aerosol size distribution is a plausible reason, the coarser particles being removed by the precipitation in the active phase. Decreases in the hygroscopicity related to decrease in the size of the particles and increase in the fine mode organic aerosols have been reported by Gunthe et al. (2009). It may be recalled that from independent measurements, Jayachandran et al. (2020b) have reported variations in the k values associated with changes in the size distribution of particles. However, this needs to be verified by more independent measurements.

The FAAM measurements provided an opportunity to examine the changes occurring in CCN characteristics in the vertical across the IGP during the contrasting phases of the monsoon. As such, we examined the CCN spectra for the free troposphere (3 to 5 km altitude) separately from the spectra for the lower altitudes (less than 3 km). This also facilitates examining the lower altitude features with those derived from the measurements aboard the Indian aircraft the pre-monsoon period, about a week to 10 days prior to phase 1 of FAAM (Jayachandran et al., 2020a), which was confined only to the lower atmosphere. The panels in Figure 9 show the results during the active phase of the monsoon, when there was a clear difference in the k

values in the upper atmosphere from those in the lower regions. It clearly emerges that over both the locations, aerosols in the upper atmosphere (free troposphere) are more hygroscopic (with lower k values) than those in the boundary layer, where the influence of local emissions would be felt more.

Vertical variations in the values of k has been examined over the Indian region based on a few aircraft measurements in the recent years under different campaigns [e.g., (Varghese et al., 2016; Jayachandran et al., 2020a)]. However, our study is the first one focusing on the transformation of CCN characteristics across the phase of the monsoon; from prior to its onset to the active phase that followed immediately. One important finding is the significant decrease in the k values (increase in the CCN activity) of aerosols across the IGP during the active phase of the monsoon, from its values just prior to the onset. Despite this feature, there is spatial distinctiveness across the IGP. In both phases of the monsoon, the central IGP with significant anthropogenic activities and associated emissions (from industries, thermal power plants, automobiles etc.) is less CCN-active with k values lying in the range (2.07 prior to onset of monsoon and 1.46 during active phase), while aerosols in the western and eastern IGP are more easily activated. Similar spatial distinctiveness has also been reported by Jayachandran et al. (2020a) during the pre-monsoon period.

Another important finding emerging from our study is the decrease in k values with altitude during the active phase of the monsoon, showing prevalence of more hygroscopic aerosols in the free troposphere. This is in sharp contrast to the results reported for the pre-monsoon period by Jayachandran et al. (2020a), who found a significant increase in k values with altitude across the entire IGP, indicating a decrease in the hygroscopicity and or increasing dominance of fine and accumulation aerosols at higher altitudes. Low values of k at higher altitudes as seen in our study are also in line with the low values reported by Dumka et al. (2015) from a Himalayan station at 2 km altitude based on measurements during the RAWEX–GVAX campaign. Similar low values of k were also reported by Roy et al. (2017) over the high altitude (2.2 km) site Darjeeling located in the eastern part of Himalayas in India. In the central peninsula Jayachandran et al. (2020b) reported smaller k values for the continental air mass compared to marine air mass due to the presence of coarser particles in the continental air mass. The near-flat CCN spectra around BBR is a consistent feature (Jayachandran et al., 2020a) and appears to be typical to coastal regions, where highly hygroscopic and coarse mode marine aerosols are available in large numbers (Jayachandran et al., 2017). However, it should be kept in mind that the k values depend on the supersaturation range used for its estimation and in our study, it was limited to only 0.4% in order to extent the measurements to higher altitudes. Similar differences between the near-surface and below-cloud values of k have also been reported by Varghese et al. (2016) during the CAIPEEX; who found higher k values (0.72) associated with polluted conditions and low k values (0.25) during clean conditions.

## 5. Summary and conclusion

Our study has brought out, perhaps for the first time over the Indian region, the contrasting features of CCN characteristics over the IGP across the pre-onset phase to the active phase of the monsoon, in the altitude region from near the surface to nearly 6 km. The salient features are:

● Prior to the onset, elevated aerosol layers prevailed throughout the IGP region, mostly above 2 km, where large increases in the number concentrations of CN and CCN were observed; though such sharp changes were not seen in the AR, except in the central IGP. The steeper aerosol spectra here with higher k values over this region suggests the prevalence of the presence of submicron aerosols with lower hygroscopicity over the central IGP. The highest CN concentrations above the boundary layer, were observed in the eastern IGP. The high wind speeds during the period, provided ideal conditions for the transport of dust from the west and anthropogenic aerosols from the central IGP (lofted by intense thermal convections) towards the eastern IGP. There existed a west to east increasing gradient in CCN concentration even above the boundary layer prior to the onset of monsoon. The lower k values over western IGP indicate the presence of coarser aerosols which are more susceptible to CCN activation. In the central peninsula, the values of CCN remained lower than those in the IGP, at all altitudes. These observations, resulted from the complex interplay of emission and advection of aerosols, along with the ABL dynamics and chemistry involving the precursor gases. There is an increase in the AR with altitude above 2 km. Compared to the freshly emitted aerosols in the boundary layer, the transported aerosols appear to be more hygroscopic.

● Strong reduction in the concentrations of CN and CCN throughout the IGP; with an east to west decreasing gradient; being most remarkable in the eastern IGP and very weak over the western IGP. This is attributed to east-west gradient (decreasing towards west) of monsoon rainfall across the IGP; with the eastern and central IGP (and the surrounding regions) receiving much higher rainfall during the active period, than the western IGP as can be seen from Figure 2 (c and d). The higher CN concentration at higher altitudes over the western IGP with values comparable or even higher than those existed prior to the onset of monsoon indicates the strong prevalence of long-range transported dust from the west, aided by the synoptic circulation, even during the active phase of the monsoon.

● During the active phase of the monsoon, the boundary layer aerosols became more hygroscopic, while the hygroscopicity of the aerosols above 3 km decreased. This appears to be caused by the change in the aerosol type after the monsoon has established. The strong monsoonal winds replaced the continental airmass that prevailed prior to the onset with moist marine airmass (figs 2a & 2b) in the lower altitudes (below 2 to 3 km). These changes can be seen in the synoptic wind at 850 hPa. The more hygroscopic aerosols present in the marine airmass increased the activation efficiency and reduced the spectral index k at the lower altitudes as seen in figures 9 (c and d) as they changed the mixing state of aerosols as has been observed by Brooks et al. (2019b). At higher altitudes, however, the mineral dust transport from the western arid regions persisted. These led to higher values of k at higher altitudes >~ 3km during the active phase.

● Consequently, the supersaturation spectrum became flatter during the active phase of the monsoon implying that aerosol will be activated at lower supersaturations. Though the local surface-based emissions (with lesser hygroscopic aerosols as seen prior to the onset of monsoon) are still active, these get mixed with the marine airmass at lower

altitudes leading to increased hygroscopicity during the active phase. Vertical lofting of surface emissions is weakened due to the weakening of the local thermal convection with the advent of monsoon and fall in temperature (by more than 10°C on an average across the IGP). As a result, the dust at higher altitudes is purer in nature and retains its less hygroscopic nature.

## Data availability

Processed data are available through the SWAAMI and parent FAAM archives at the Centre for Environmental Data Analysis (CEDA) (http://data.ceda.ac.uk/badc/faam, last access: xxxx).

## Author contribution

SKS, KKM and HC together conceived of the experiment; MRM, JT and HC participated in the field campaign. JT collected the data and performed the quality check. MRM carried out the scientific data analysis and prepared the draft of the
manuscript. KKM and SKS were involved in the scientific interpretation of the results, leading to the formulation of the manuscript, and along with HC and JT reviewed and revised the manuscript.

## Acknowledgements

A number of institutions were involved in logistics, planning, and support of the campaign: the Indian Institute of Science, Vikram Sarabhai Space Centre, University of Reading and the Met Office, UK. We gratefully acknowledge support from all.
Airborne data was obtained using the BAe-146-301 Atmospheric Research Aircraft flown by Airtask Ltd and managed by FAAM Airborne Laboratory, jointly operated by UKRI and the University of Leeds. We thank Divecha Centre for Climate Change for the support. ERA-Interim wind field data were provided courtesy of ECMWF. The gridded rainfall data was provided courtesy of ECMWF. We also thank the NOAA Air Resources Laboratory (ARL) for the provision of the HYSPLI transport and dispersion model used in this publication. Sreedharan Krishnakumari Satheesh acknowledges the support of the
JC Bose Fellowship from the Department of Science and Technology, New Delhi.

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

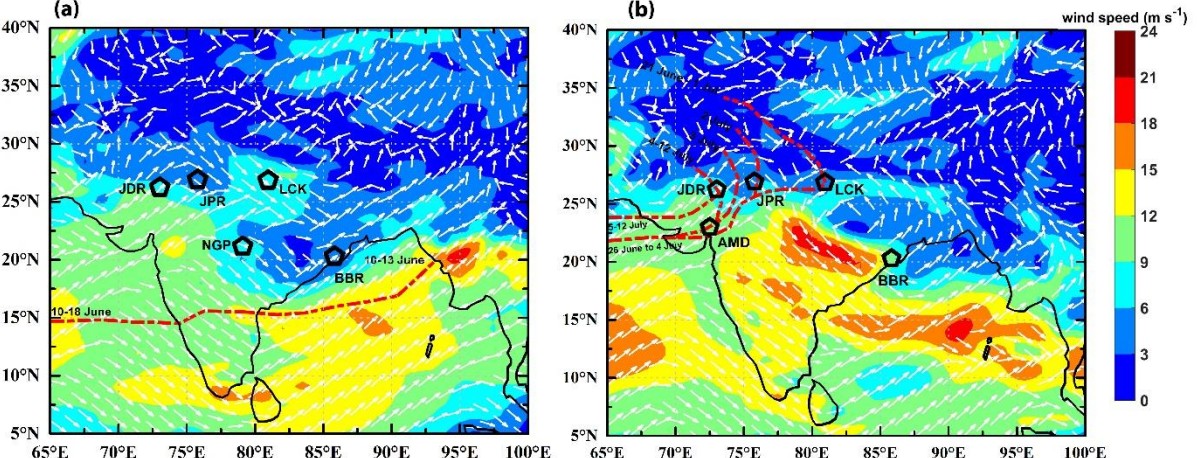

**Figure 1: The advance of monsoon and the synoptic wind field at 850 hPa during a) phase 1 and b) phase 3 of the SWAAMI campaign. The red dotted line indicates the northern limit of monsoon.**


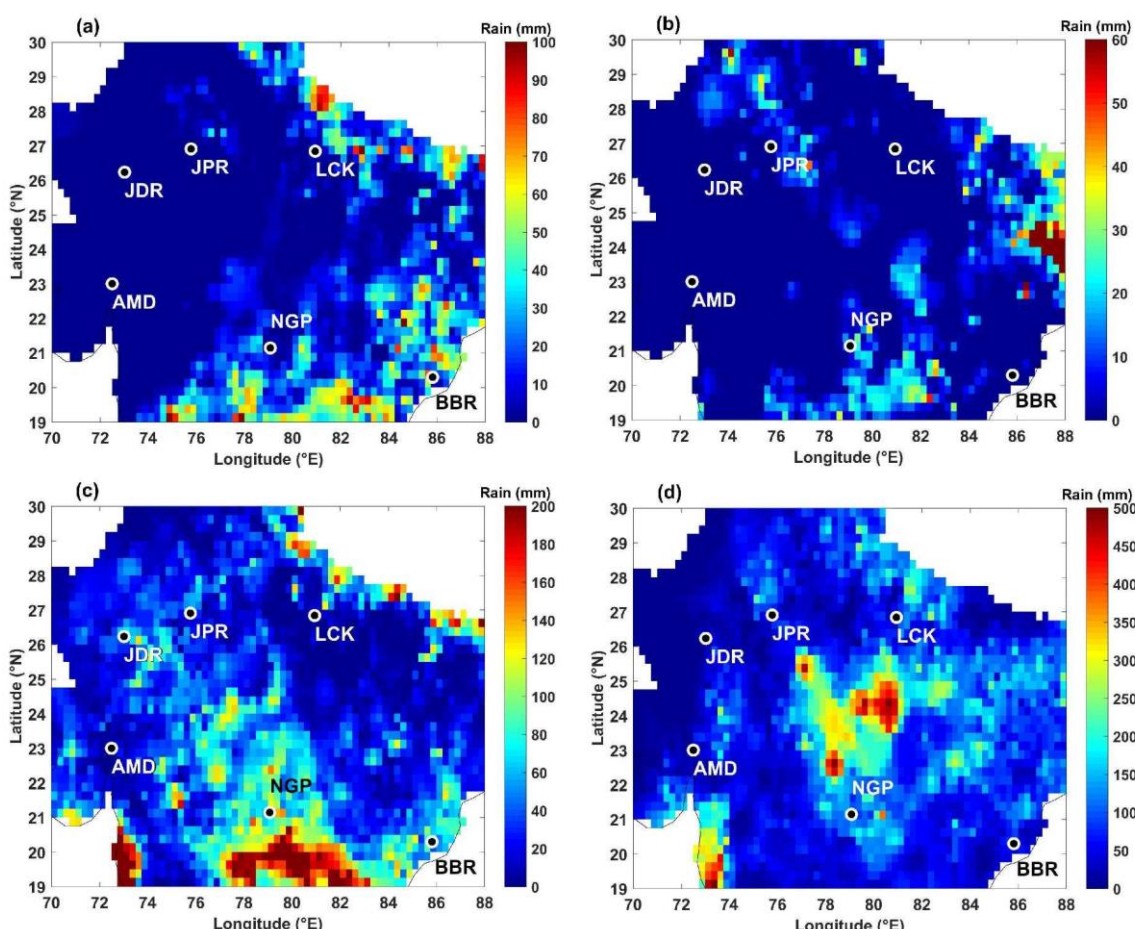

**Figure 2: Accumulated rainfall for the periods a) 4-10 June (1 week prior to phase 1 measurements) b) 11-13 June (during phase 1 measurements) c) 25-June to 1-July (1 week prior to phase 3 measurements) and d) 2-7 July (during phase 3 measurements of the SWAAMI campaign).**


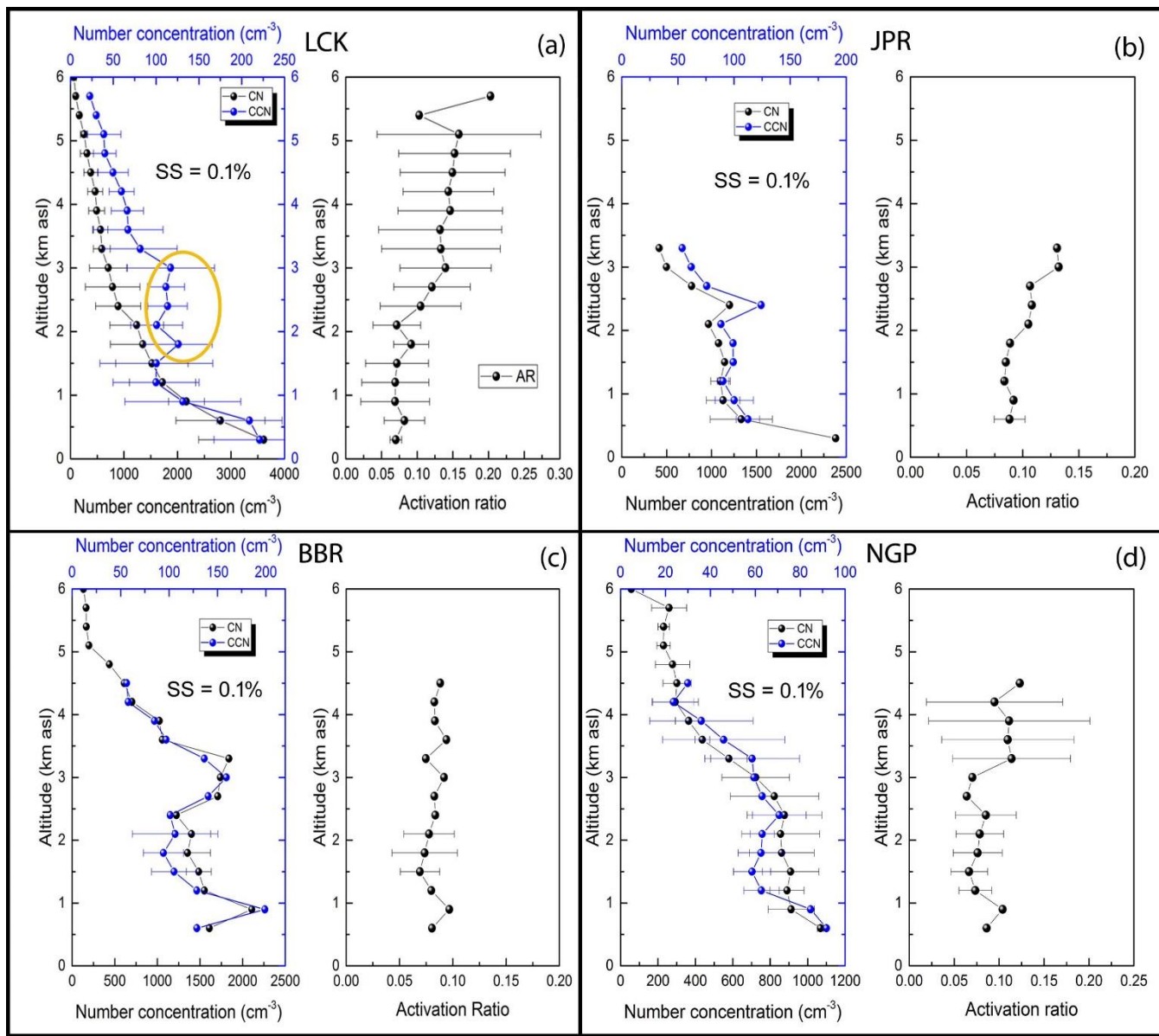

**Figure 3:** The number concentrations of the CN, CCN at 0.1% supersaturation and AR over LCK, JPR, BBR and NGP are shown in figures 3a, 3b, 3c and 3d respectively. The activation ratio gives the fraction of the total aerosols which can be converted to CCN at 0.1% supersaturation.


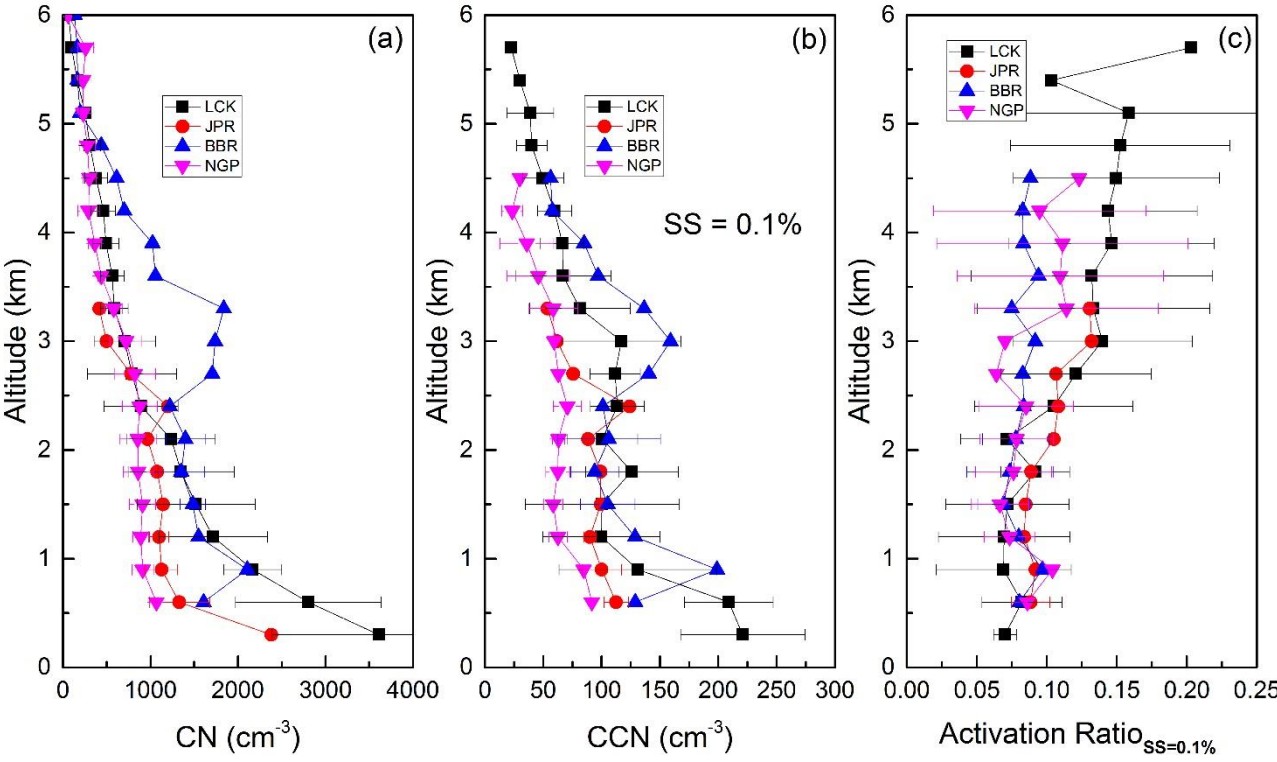


**Figure 4: Comparison of CN, CCN and AR values at the different locations during phase 1 of the campaign. The CCN and AR corresponding to 0.1% supersaturation are shown.**

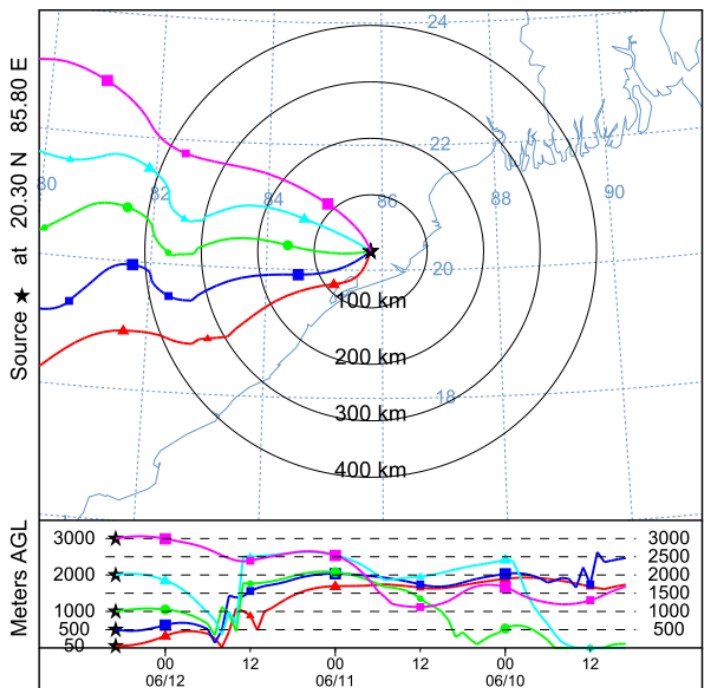

**Figure 5: 3-days back trajectories of the air mass arriving at BBR at altitudes 50 m, 500 m, 1 km, 2 km and 3 km during our measurements.**

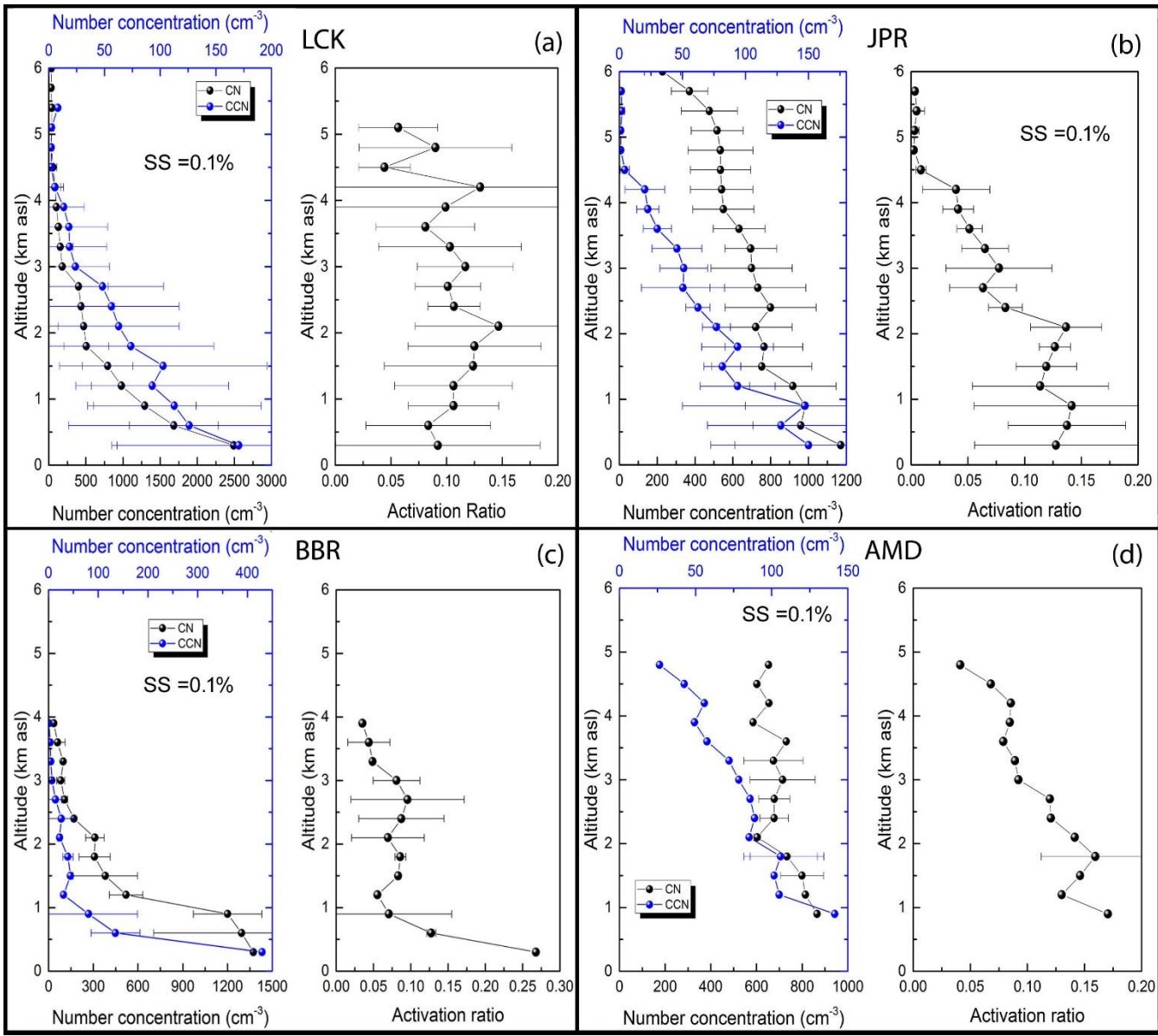

**Figure 6: The number concentrations of the CN, CCN and AR over LCK, JPR, BBR and AMD are shown in panels a, b, c and d respectively. The activation ratio represents the fraction of total aerosols which can be converted to CCN at 0.1% supersaturation.**


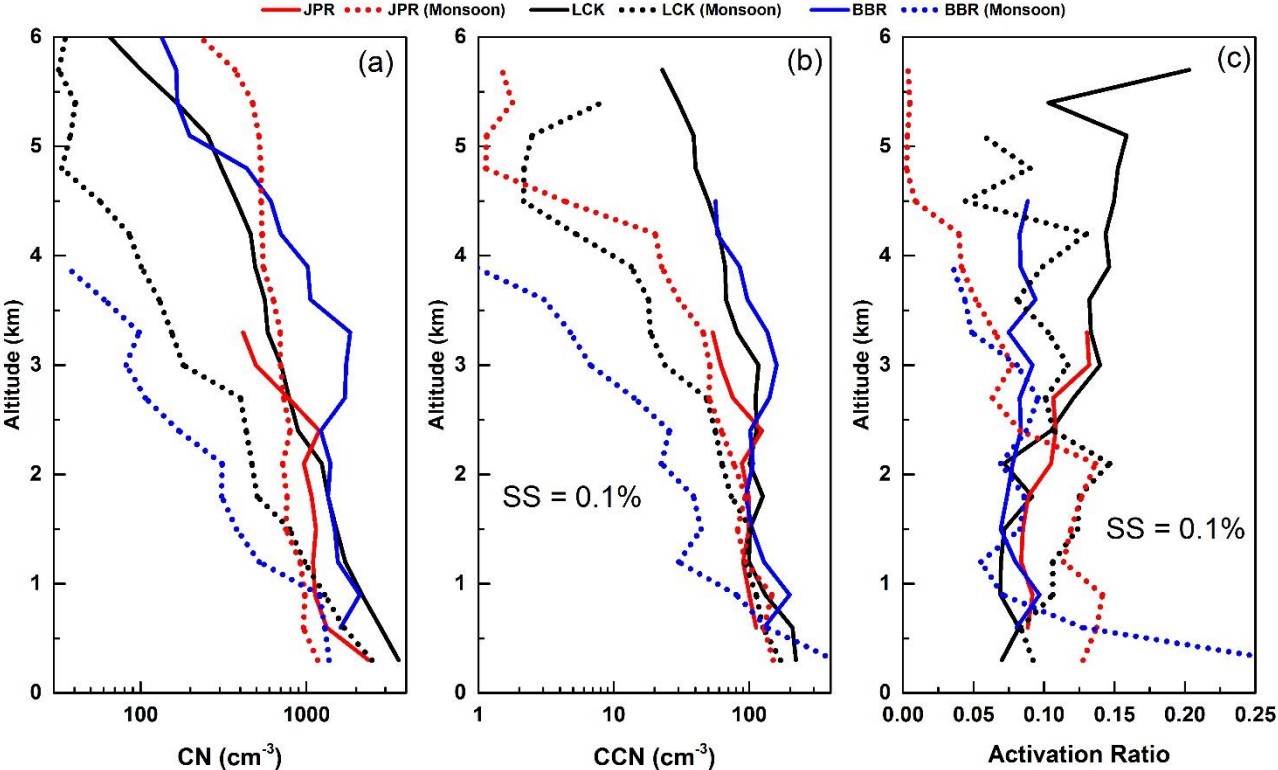

**Figure 7: The changes in the altitude variations of a) CN concentrations, b) CCN concentrations and c) activation ratios during the SWAAMI campaign. The solid lines represent the values during phase 1 and the dotted lines show the values during phase 3.**


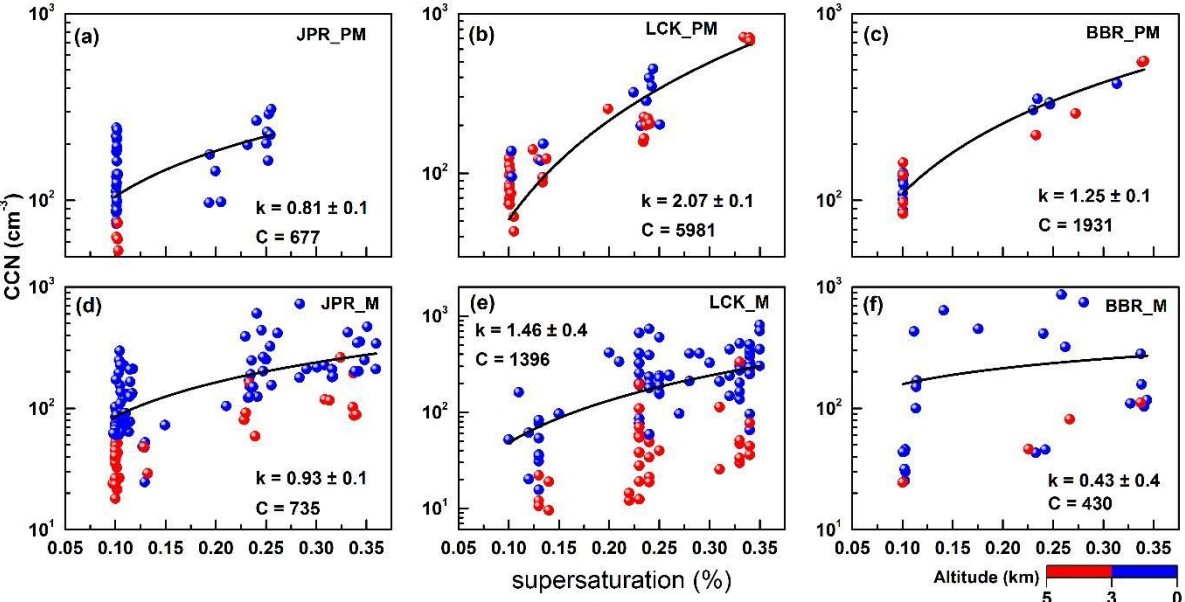

**Figure 8: Spatial variation of CCN spectra across the IGP for the western (JPR), central (LCK) and eastern (BBR) IGP regions; top panels (a, b & c) represent the scenarios prior to the onset of monsoon while bottom panels (d, e & f) correspond to the active monsoon phase. The points are individual measurements; blue colour stands for lower altitudes (<3km) and red colour represents the free tropospheric measurements (3 to 5 km altitudes). The lines are regression fit to equation (1).**


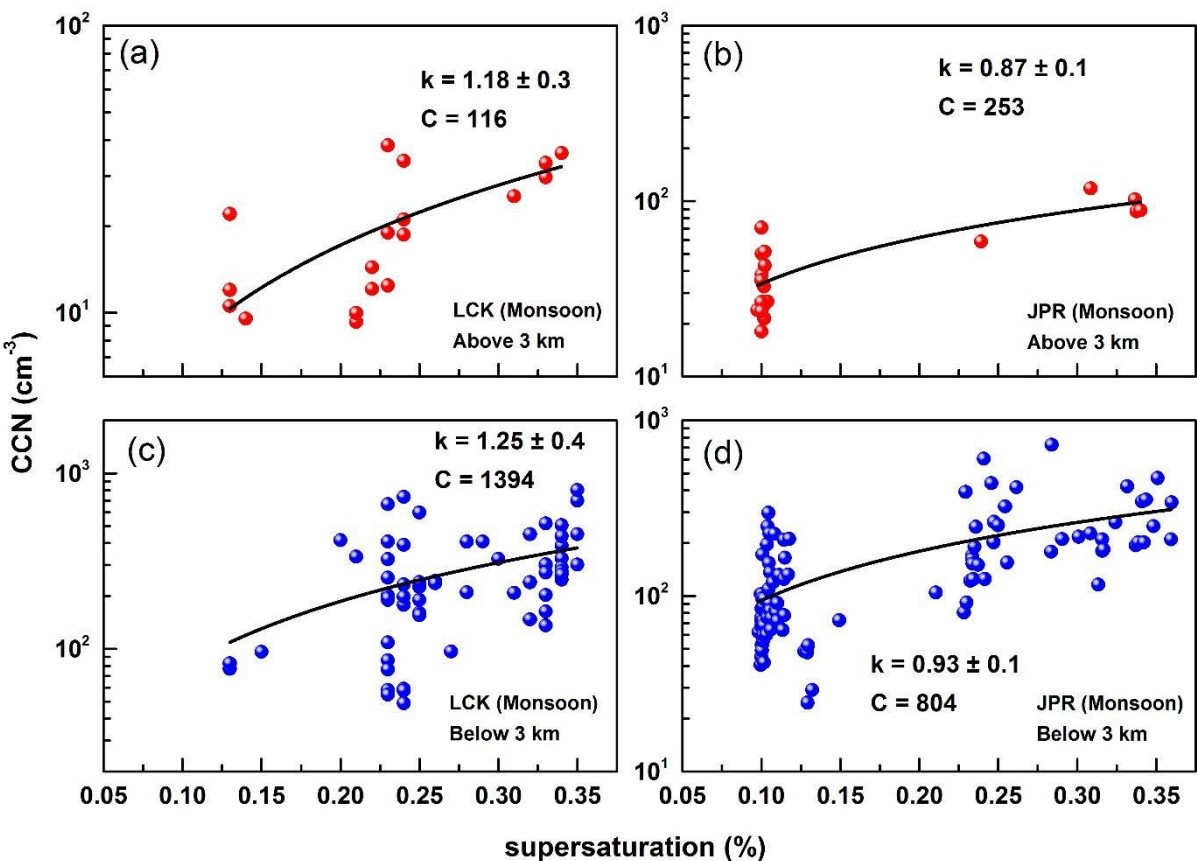

**Figure 9: Vertical variation of CCN spectra during active phase of the monsoon: Top panels show the features in the free troposphere (3 to 5 km) and bottom panels for the lower altitude region (below 3 km). Panels on the left are for central IGP and those on the right for western IGP. The measurements over BBR was carried out only on a single day and the concentrations of CN and CCN sharply dropped above 3 km limiting the availability of data above this altitude. Hence the change in k above and below 3 km is not included in the discussion.**



**Table 1: Details of measurements**


| Sl. No. | DATE | Phase | Flight Code | Section | Region Covered | Track | Location |
|---------|------|-------|-------------|---------|----------------|-------|----------|
| 1 | 11[th] June | 1 | B956 | A | LCK to JPR | West of LCK | 80.3 ±0.3, 27.1 ±0.2 |
| | | | | B | | West of JPR | 75.3 ±0.0, 26.5 ±0.1 |
| | | | | C | | East to west JPR | 76.9 ±0.8, 27.6 ±0.1 |
| 2 | 12[th] June | 1 | B957 | A | LCK to BBR | East of LCK | 81.2 ±0.3, 26.4 ±0.2 |
| | | | | B | | BBR | 85.6 ±0.1, 20.4 ±0.1 |
| | | | | C | | Northwest of BBR | 83.8 ±0.6, 22.9 ±1.3 |
| | | | | D | | East of LCK | 83.0 ±0.0, 25.0 ±0.0 |
| | | | | E | | East of LCK | 81.1 ±0.1, 26.5 ±0.1 |
| 3 | 13[th] June | 1 | B958 | A | NGP to BLR | South of LCK | 80.5 ±0.1, 26.4 ±0.2 |
| | | | | B | | NGP | 78.8 ±0.1, 20.6 ±0.3 |
| | | | | C | | South of NGP | 79.2 ±0.2, 20.2 ±0.0 |
| 4 | 2[nd] July | 3 | B969 | A | LCK to JPR | LCK | 80.5 ±0.3, 27.1 ±0.2 |
| | | | | B | | JDR | 73.9 ±0.7, 26.4 ±0.1 |
| | | | | C | | JDR to JPR | 76.3 ±1.5, 27.1 ±0.4 |
| | | | | E | | LCK | 80.8 ±0.2, 26.7 ±0.0 |
| 5 | 3[rd] July | 3 | B970 | A | LCK to JPR | LCK | 80.3 ±0.3, 27.1 ±0.2 |
| | | | | B | | JDR | 72.9 ±0.0, 26.1 ±0.0 |
| | | | | C | | JDR to JPR | 76.3 ±1.8, 27.0 ±0.5 |
| | | | | D | | West of LCK | 79.4 ±0.2, 27.3 ±0.0 |
| | | | | E | | LCK | 80.7 ±0.2, 26.8 ±0.0 |
| 6 | 4[th] July | 3 | B971 | A | LCK to BBR | LCK | 81.3 ±0.3, 26.4 ±0.2 |
| | | | | B | | BBR | 86.1 ±0.1, 19.8 ±0.1 |
| | | | | C | | BBR to LCK | 85.2 ±0.7, 20.8 ±0.7 |
| | | | | D | | BBR to LCK | 83.5 ±0.2, 23.1 ±0.7 |
| | | | | E | | LCK | 81.2 ±0.1, 26.5 ±0.1 |
| 7 | 5[th] July | 3 | B972 | A | LCK to JPR | LCK | 80.3 ±0.3, 27.2 ±0.2 |
| | | | | B | | JDR | 73.8 ±0.8, 26.3 ±0.2 |
| | | | | C | | JDR to JPR | 74.5 ±0.8, 26.5 ±0.1 |
| | | | | D | | North of JPR | 76.4 ±0.3, 27.1 ±0.2 |
| | | | | E | | West of LCK | 79.1 ±0.6, 27.4 ±0.2 |
| | | | | F | | LCK | 80.5 ±0.2, 26.8 ±0.1 |
| 8 | 6[th] July | 3 | B973 | A | LCK to JPR | LCK | 80.3 ±0.4, 27.2 ±0.5 |
| | | | | B | | JPR | 75.3 ±0.0, 26.8 ±0.2 |
| | | | | C | | South of JPR | 75.3 ±0.0, 26.3 ±0.3 |
| | | | | D | | JPR | 75.4 ±0.0, 25.9 ±0.1 |
| | | | | E | | East of JPR | 76.4 ±0.3, 26.9 ±0.0 |
| | | | | F | | West of LCK | 79.5 ±0.9, 27.0 ±0.2 |

| 9 | 7th July | 3 | B974 | A | LCK to AMD | LCK | 80.7 ±0.1, 26.2 ±0.3 |
|---|---|---|---|---|---|---|---|
| | | | | B | | SE of AMD | 74.8 ±0.3, 22.4 ±0.1 |
| | | | | C | | East of AMD | 76.4 ±0.3, 22.9 ±0.1 |
| | | | | D | | East of AMD | 77.0 ±0.1, 23.1 ±0.0 |
| | | | | F | | LCK | 80.7 ±0.0, 26.6 ±0.1 |