# Peer review of "Measurement report: Altitudinal variation of CCN activation across the Indo-Gangetic Plains prior to monsoon onset and during peak monsoon periods: Results from the SWAAMI field campaign"

_Atmospheric Chemistry and Physics, 2020_

## Referee Comment (RC1) · Anonymous Referee #1 · 20 Jan 2021

General Comments This study investigates the vertical distributions (altitude profiles) of the condensation nuclei (CN) and cloud condensation nuclei (CCN), based on airborne measurements during the SWAAMI field campaign (June to July, 2016) over the Indo-Gangetic Plains (IGP). The paper examines the vertical distributions of CN, CCN and the activation ratio (AR=CCN/CN) and compares the changes on them between locations in east and west Indo-Gangetic Plains (IGP) as well as between the premonsoon and monsoon season in order to assess the role of different aerosol types

and chemical composition on CCN activation just prior and during the Indian summer monsoon. It's an interesting work, well within the scope of the journal and useful for scientists of the atmospheric aerosol community. This paper comes as a continuation of previous ones (by the same team) regarding aerosol properties, radiative effects and compositions during the SWAAMI campaign. The paper is well organized and the narrative flows well, however, some more discussions and references at specific parts that will enhance the importance of the current results and will provide physical explanations and comparisons with previous studies are needed. Specific Comments In the Introduction section, authors should insist more on previous works in India dealing with CCN, since they practically refer only to CAIPEEX campaign (airborne measurements). Looking the literature, I can see several other works with ground instrumentation as well, some of them also cited later in the manuscript like Dumka et al. (2015), Jayachandran et al. (20020a, 2020b), etc.. You may also see Jayachandran, et al., 2018 (Atm.Res.), Dipu et al. (2013, AtmEnv), Bhattu et al. (2014, AtmEnv)... Line 83: BBR is not located in the eastern IGP, but at the eastern Indian coast. This should be mentioned here, since reader has a wrong thought about the geographical distribution of the sites. Also, you should clarify that JPR is at the arid zone and is not also considered as a standard IGP site. Line 114: The reference Dee et al. (2011) should accompany the meteo dataset used i.e. ERA-Interim or ERA-5 reanalysis. So, this should be referred here. Lines 145-147: I would expect a more detailed discussion here related with aerosol hygroscopicity and types over the IGP region. Obviously, aerosol chemical properties play the major role in aerosol hygroscopicity and CCN activation. These chemical properties, as well as aerosol types should be discussed here, in view of changes in CCN and hygroscopicity. Literature, global and Indian, are rich in this issue and should be used here. Line 150: Correct as CN and CCN Line 159: There is only marginal differences in the AR compared to Lucknow. I think that there is no important difference between these AR values deserving further explanation from the view of the physical and chemical aerosol point of view. So, authors should refer to marginal or slight differences between these sites, at least for the AR values near

the surface, indicating low aerosol hygroscopicity. Lines 169-170: This difference in aerosol types also exists in the vertical between west and east IGP and this should be further discussed in the manuscript, along with dominant aerosol types and chemical compositions between the regions. Line 221: The role of the water-soluble organics should be highlighted here. Lines 127-248: References are needed here. Lines 249-251: A recent work at the Indian Himalayas (Nainital site, 1958 m; Dumka et al., 2021, STOTEN), which classified the aerosol types based on in situ surface observations showed that the fresh BC aerosols of local origin (BC-dominated type) was much less hygroscopic than the coated and aged "large-BC type", which was mostly transported by the IGP. This finding supports the current results and should be an advance in the discussions about different hygroscopicity levels from various aerosol types in India. Lines 255-257: This statement is true and also supported by previous studies that show an increase in water-vapor content (from AERONET) during pre-monsoon dust events over the IGP (Prasad and Singh, 2007, JGR; Sarvan Kumar et al., 2015, Aeolian Research, etc), indicating a mixing of marine-dust air masses. In addition, more recently, Dumka et al. (2019, JGR) showed that the dust emissions and dust-storm propagation over the Thar desert in pre-monsoon is highly controlled by the SW monsoon density currents over the land area, which may increase the WVC, and therefore, the aerosol hygroscopicity. Lines 340-367: In the major findings discussed here, you may increase the literature overview about aerosol size and chemical properties. Line 345: Delete "IGP". Line 366: Delete "of". Figure 1: The experimental sites should be clearly visible in Fig. 1. Increase the fonts, make the sites clearly visible. Figure 6 Caption. Correct the figure numbers there.
* * *

---

## Referee Comment (RC2) · Anonymous Referee #2 · 2 Mar 2021

The study represents vertical profiles of CN and CCN particles and their spatial variability across the Indo-Gangetic Plain in order to capture differences prior to the monsoon and during the monsoon. Differences in levels and activation ratios are observed and linked to possible different aerosol composition/hygroscopicity. Differences are also observed between different altitudes, with pronounced changes namely above 3 km.

The study is interesting but several important issues have to be addressed before the

manuscript is considered for further publication.

General comments:

1) A more thorough review of relevant literature in the area and on the subject should be presented. Many recent studies are not mentioned, and what the recent study offers in comparison to others is not clear.

2) There is a complete lack of mentioning operational supersaturation levels, which is crucial for a notion of particle activation size. Without this information all discussion falls short. Also other sampling information, such as drying of the aerosol prior to CCN and CN measurement should be mentioned.

Specific comments:

Introduction: As the manuscript refers to CCN and hygroscopicity, the importance of chemical composition should be also discussed and a more excessive review of the literature in the area should mentioned. To my knowledge, there are at least two recent studies focusing on CCN in the area, also taking into account chemical composition and number size distribution. Shika et al. (2020) focus on aerosol properties also during pre-monsoon and during monsoon season and implications on cloud droplet formation. Furthermore, Arub et al. (2020) characterize chemical composition and size distributions in the area of Delhi based on air masses origin and their impact on hygroscopicity and CCN formation. Singla et al. (2017) study the role of organics in CCN activation in Western Ghats, India. Also another study part of the same experiment (CAIPEEX) by Jayachandran et al. (2020a) although mentioned in the discussion section (4.1.1) general outcomes are not mentioned in the introduction, in order to put into context the present study. Finally, Jayachandran et al. (2020b) also report airborne CCN measurements across the Indo-Gangetic Plain which also are mentioned in the discussion section (4.1.1) but not mentioned in the introduction. Overall, the introduction section needs to be enriched with other relevant studies in the area so that the current study is put into context.

[Figure]

P3, L80-86: A map with the locations of the focus areas would be helpful for the reader to get an idea of the topography and type of environment and possible aerosol sources which can impact aerosol size and chemical composition.

P4, L100-108: Is there a drier at the inlet? What is the RH of the sampled aerosol which enters inside the CCN counter and the CPC? Also how was the CCN instrument operated? Was it on scanning flow analysis (Moore and Nenes, 2009; Moore et al., 2012; Lathem et al., 2013)? To my knowledge, this is the most appropriate analysis for airborne measurements as it ensures the correct supersaturation spectra over very limited timescales. If not, the CCN analysis by staying at a constant supersaturation for a given time allows for a complete CCN spectrum every, say, hour, during which obviously the aircraft has moved on to other areas, with other aerosol characteristics and sources. Even in Trembath (2013) it is not clearly stated how the CCN instrument supersaturation varied:

"Each column supersaturation was set using the proprietary dual columnCCN software (DMT inc, Boulder); the set point ranged between 0.1 and 0.5 % across all flights." p.122

Or was the CCN instrument operated in a constant supersaturation? A few details on the operating mode should be included, and how the time at each supersaturation compares in terms of aircraft velocity and distance covered.

P5, L136-148, Figures 3 & 4: Are all provided CCN and AFs at 0.1% supersaturation? If yes, it should be clear both in the figures and the text. There is no mention whatsoever of the instrument supersaturation in the text, not at what particle sizes this supersaturation corresponds to.

P6, L168-174: Once more, no mention of instrument supersaturation. Was it constant? Was it the same during all flights for which the ARs are compared between sites?

P6, L181-184: When evoking the anthropogenic impact, anthropogenically impacted

emissions are mostly in the lower particle sizes, which means that particles indeed activate in lower ranges of supersaturation. Once more the instrument supersaturation and respective particle size range should be clearly stated.

P7, L203-214: All this discussion should be put in context also with particle size. Sulfate is mostly found in particle sizes larger than organics.

P7, L215-225: Therefore the current study offers insight of what happens above the boundary layer? This is the difference between the other studies (Brooks et al., 2019a; Jayachandran et al., 2020a)? This should be clarified, even in the introduction section.

P10, L294-195: Operational mode and settings should be comparable to those during the prior monsoon period, correct? Otherwise no comparison is possible.

Figures 8 & 9: It is clear from these figures that the CCN instrument was operated in different supersaturation levels, therefore it becomes even more imperative that the whole discussion on ARs is clarified, as well as operational conditions between pre-monsoon and monsoon flights. Also the scatter in these figures is sometimes so high, which raises confidence issues concerning the fitting (e.g. Fig. 8 d & e, 9c)

Technical corrections

P7, L218: precursor gases (one word)

References

Shika, S., Gadhavi, H., Suman, M.N.S. et al. Atmospheric aerosol properties at a semi-rural location in southern India: particle size distributions and implications for cloud droplet formation. SN Appl. Sci. 2, 1007 (2020). https://doi.org/10.1007/s42452-020-2804-2

V. Singla, S. Mukherjee, P.D. Safai, G.S. Meena, K.K. Dani, G. Pandithurai: Role of organic aerosols in CCN activation and closure over a rural background site in Western Ghats, India, Atmospheric Environment, Volume 158, 148-159,

https://doi.org/10.1016/j.atmosenv.2017.03.037, 2017.

Arub, Z., Bhandari, S., Gani, S., Apte, J. S., Hildebrandt Ruiz, L., and Habib, G.: Air mass physiochemical characteristics over New Delhi: impacts on aerosol hygroscopicity and cloud condensation nuclei (CCN) formation, Atmos. Chem. Phys., 20, 6953–6971, https://doi.org/10.5194/acp-20-6953-2020, 2020.

Jayachandran, V. N., Varghese, M., Murugavel, P., Todekar, K. S., Bankar, S. P., Malap, N., Dinesh, G., Safai, P. D., Rao, J., Konwar, M., Dixit, S., and Prabha, T. V.: Cloud condensation nuclei characteristics during the Indian summer monsoon over a rain-shadow region, Atmos. Chem. Phys., 20, 7307–7334, https://doi.org/10.5194/acp-20-7307-2020, 2020a.

Jayachandran, V. N., Suresh Babu, S. N., Vaishya, A., Gogoi, M. M., Nair, V. S., Satheesh, S. K., and Krishna Moorthy, K.: Altitude profiles of cloud condensation nuclei characteristics across the Indo-Gangetic Plain prior to the onset of the Indian summer monsoon, Atmos. Chem. Phys., 20, 561–576, https://doi.org/10.5194/acp-20-561-2020, 2020b.

Richard H. Moore & Athanasios Nenes (2009) Scanning Flow CCNAnalysis—A Method for Fast Measurements of CCN Spectra, Aerosol Science and Technology,43:12, 1192-1207, DOI: 10.1080/02786820903289780.

Moore, R. H., Cerully, K., Bahreini, R., Brock, C. A., Middlebrook, A. M., and Nenes, A. (2012), Hygroscopicity and composition of California CCN during summer 2010, J. Geophys. Res., 117, D00V12, doi:10.1029/2011JD017352.

Lathem, T. L., Beyersdorf, A. J., Thornhill, K. L., Winstead, E. L., Cubison, M. J., Hecobian, A., Jimenez, J. L., Weber, R. J., Anderson, B. E., and Nenes, A.: Analysis of CCN activity of Arctic aerosol and Canadian biomass burning during summer 2008, Atmos. Chem. Phys., 13, 2735–2756, https://doi.org/10.5194/acp-13-2735-2013, 2013.

---

## Author Comment (AC1) · 22 Apr 2021

At the outset, we thank the reviewer for the meticulous review, constructive comments and the overall appreciation of the work. We have considered each of the comments carefully and revised the manuscript. Our responses to the comments, which formed the basis for the revision, are given below along with the page and line numbers in the

revised manuscript, where the revisions are incorporated.

Reply to comments of Anonymous Reviewer #1 (RC1)

1) In the Introduction section, authors should insist more on previous works in India dealing with CCN, since they practically refer only to CAIPEEX campaign (airborne measurements). Looking the literature, I can see several other works with ground instrumentation as well, some of them also cited later in the manuscript like Dumka et al. (2015), Jayachandran et al. (20020a, 2020b), etc.. You may also see Jayachandran, et al., 2018 (Atm.Res.), Dipu et al. (2013, AtmEnv), Bhattu et al. (2014, AtmEnv). . .

Yes. There have been several works with CCN in India, in addition to those cited by the reviewer. However, most of these studies were ground-based focusing on case studies or long-term measurements on seasonality. However, we focus on the vertical structure of CCN characteristics within the ABL and in the free troposphere and the changes that occur in these properties as the season changes from just prior to the onset of Indian monsoon to its active phase. There are very few studies, if not none, dealing with such details and addressing to the changes in the aerosol characteristics leading to these changes. Accordingly, our measurements did not cover the near surface CN and CCN properties. As such, we have cited only those studies dealing with vertical structure of CN/CCN properties and that explains why our references have been limited to airborne measurements. As it emerges in our study, the vertical structure of CCN characteristics significantly differs from the surface and also vary with the phase of the monsoon. Nevertheless, as suggested by the reviewer, we have cited a few ground-based measurements that address to at least a few of the features examined in this study (page 2, lines 48-49; page 2-3, lines 56-79 and page 3, lines 87-88).

2) Line 83: BBR is not located in the eastern IGP, but at the eastern Indian coast. This should be mentioned here, since reader has a wrong thought about the geographical distribution of the sites. Also, you should clarify that JPR is at the arid zone and is not also considered as a standard IGP site.

We agree partly. While Bhubaneswar is located just south of the eastern boundary of the IGP. Jodhpur and Jaipur are located at the southern tip of the western IGP, even though they exhibit arid characteristics, and are chosen for the convenience of aircraft operations. Furthermore, the sorties towards and from these base stations have been mostly through the core IGP region. The Indo-Gangetic Plain (IGP), also known as the Indus-Ganga Plain and the North Indian River Plain, is a 2.5-million km2 fertile plain encompassing northern regions of the Indian subcontinent, including most of northern and eastern India, the eastern parts of Pakistan, virtually all of Bangladesh and the southern plains of Nepal. The region is named after the Indus and the Ganges rivers and encompasses a number of large urban areas. The plain is bound on the north by the Himalayas, which feed its numerous rivers and are the source of the fertile alluvium deposited across the region by the two river systems. The southern edge of the plain is marked by the Chota Nagpur Plateau. On the west rises the Iranian Plateau (source Wikipedia). While representing the IGP, it is often observed that the Rajasthan Plain and the southern part of the upper Ganga Plain are excluded because of the arid or semi-arid nature. We have provided a figure, that shows the geographical extent of the IGP (IGP is not confined to India, it is defined by the rivers Indus, Ganges and Brahmaputra as detailed below) and the base stations (https://commons.wikimedia.org/w/index.php?curid=2454397). As per the broad definition of IGP, JDR is part of the Rajasthan plain and Jaipur is located very close to the Punjab Haryana plain and upper Ganga plain and surrounded by the plains on three sides. Both Jaipur and Jodhpur are located in semi-arid parts of western India. The figure and explanations clarify this beyond doubt. We have revised the manuscript accordingly (page 4, lines 112-113). The map of the IGP was taken from https://commons.wikimedia.org/w/index.php?curid=2454397. Last accessed on 16-Feb-2021.

3) Line 114: The reference Dee et al. (2011) should accompany the meteo dataset used i.e. ERA-Interim or ERA-5 reanalysis. So, this should be referred here.

Complied with in the revised manuscript (page 5, lines 149-151).

4) Lines 145-147: I would expect a more detailed discussion here related with aerosol hygroscopicity and types over the IGP region. Obviously, aerosol chemical properties play the major role in aerosol hygroscopicity and CCN activation. These chemical properties, as well as aerosol types should be discussed here, in view of changes in CCN and hygroscopicity. Literature, global and Indian, are rich in this issue and should be used here.

This is an important suggestion. We draw the attention of the reviewer to the discussion already provided on the importance of hygroscopicity, aerosol types and size distribution (pages 8-9, lines 248-289). We have also added a few more references in pages 7 (lines 216-219) and page 8 (lines 249-251).

5) Line 150: Correct as CN and CCN

Corrected in the revised manuscript (page 6, line 192).

6) Line 159: There is only marginal differences in the AR compared to Lucknow. I think that there is no important difference between these AR values deserving further explanation from the view of the physical and chemical aerosol point of view. So, authors should refer to marginal or slight differences between these sites, at least for the AR values near the surface, indicating low aerosol hygroscopicity.

It is true that there is only a marginal difference in the AR between Lucknow and Bhubaneswar. But, what makes it worth discussing further is that, this difference arise due to the changes in the vertical distribution of the CCN concentration, while that of the CN remained well comparable at Lucknow and Bhubaneswar.

7) Lines 169-170: This difference in aerosol types also exists in the vertical between west and east IGP and this should be further discussed in the manuscript, along with dominant aerosol types and chemical compositions between the regions.

Yes, the reviewer is absolutely right, and this is an important outcome. We have discussed the changes in aerosol types with altitude across the IGP and the main points are highlighted in the conclusions. This study does not explore the chemical composition (as these were not determined) but concurrent measurements of chemical composition are reported by Brooks et al. (2019), which is referred wherever necessary to strengthen our arguments.

8) Line 221: The role of the water-soluble organics should be highlighted here.

We did not find that this is required here, because the main point is the reduction in activation efficiency due to enhancement in BC concentration; this adequately takes care of the observations.

9) Lines 127-248: References are needed here.

We re-examined this portion and found that most of the important and relevant works are cited. However we have added more references explaining the role of changes in chemical composition and size distribution of aerosols on the CCN activation (page6, lines 173-176, 188-189; page 7, lines 216-219; page 8, lines 249-251; pages 10, lines 298-300).

10) Lines 249- 251: A recent work at the Indian Himalayas (Nainital site, 1958 m; Dumka et al., 2021, STOTEN), which classified the aerosol types based on in situ surface observations showed that the fresh BC aerosols of local origin (BC-dominated type) was much less hygroscopic than the coated and aged "large-BC type", which was mostly transported by the IGP. This finding supports the current results and should be an advance in the discussions about different hygroscopicity levels from various aerosol types in India.

Yes, we agree. We have cited this work and a related work in the revised manuscript (pages 10, lines 298-299).

11) Lines 255-257: This statement is true and also supported by previous studies that show an increase in water-vapor content (from AERONET) during pre-monsoon

dust events over the IGP (Prasad and Singh, 2007, JGR; Sarvan Kumar et al., 2015, Aeolian Research, etc), indicating a mixing of marine-dust air masses. In addition, more recently, Dumka et al. (2019, JGR) showed that the dust emissions and dust-storm propagation over the Thar desert in pre-monsoon is highly controlled by the SW monsoon density currents over the land area, which may increase the WVC, and therefore, the aerosol hygroscopicity.

Yes, the reviewer is correct. However, our point here is the gathering of moisture by the airmass passing over ocean. Furthermore, we have not encountered any dust storm or associated changes in columnar water vapour during our measurements.

12) Lines 340-367: In the major findings discussed here, you may increase the literature overview about aerosol size and chemical properties.

Complied with in the revised manuscript (page 13, line 394-395, 403, and 416-418).

13) Line 345: Delete "IGP".

Corrected in the revised manuscript.

14) Line 366: Delete "of".

Corrected in the revised manuscript.

15) Figure 1: The experimental sites should be clearly visible in Fig. 1. Increase the fonts, make the sites clearly visible.

Figure 1 has been modified in the revised manuscript.

16) Figure 6: Caption. Correct the figure numbers there.

Corrected in the revised manuscript.

We thank the reviewer for the detailed comments.

References Brooks, J., Allan, J.D., Williams, P.I., Liu, D., Fox, C., Haywood, J., Langridge, J.M., Highwood, E.J., Kompalli, S.K., O'Sullivan, D., Babu, S.S., Satheesh,

S.K., Turner, A.G., Coe, H., 2019. Vertical and horizontal distribution of submicron aerosol chemical composition and physical characteristics across northern India during pre-monsoon and monsoon seasons. Atmos. Chem. Phys. 19, 5615-5634.

Please also note the supplement to this comment:
https://acp.copernicus.org/preprints/acp-2020-1233/acp-2020-1233-AC1-supplement.pdf

[Figure]

**Fig. 1.** Figure 1. The Indo-Gangetic Plain is represented by the shaded (blue-magenta) region. The measurement sites are represented by the circles.

**Supplement:**

**Measurement report: Altitudinal variation of CCN activation across the Indo-Gangetic Plains prior to monsoon onset and during peak monsoon periods: Results from the SWAAMI field campaign**

[Figure]

**Figure S1: Flight track during phase 1 measurements**

[Figure]

**Figure S2: Flight track during phase 3 measurements**

[Figure]

**Figure S3: 96 hour back trajectory of air mass arriving at JPR on 11th June at altitudes 50 m, 500 m, 1 km, 2 km and 3 km.**

[Figure]

**Figure S4: 72 hour back trajectory of air mass arriving at LCK on 11th June at altitudes 50 m, 500 m, 1 km, 2 km and 3 km.**

**NOAA HYSPLIT MODEL**
**Backward trajectories ending at 0700 UTC 12 Jun 16**
**CDC1 Meteorological Data**

[Figure]

[Figure]

**Figure S5: 72 hour back trajectory of air mass arriving at LCK on 12th June at altitudes 50 m, 500 m, 1 km, 2 km and 3 km.**

[Figure]

**Figure S6: 72 hour back trajectory of air mass arriving at LCK on 13th June at altitudes 50 m, 500 m, 1 km, 2 km and 3 km.**

[Figure]

**Figure S7: 72 hour back trajectory of air mass arriving at AMD on 7th July at altitudes 50 m, 500 m, 1 km, 2 km and 3 km.**

**Sampling of ambient aerosols during the campaign**

The Condensation Nuclei (CN) concentration was estimated using a modified water filled Condensation Particle Counter (CPC) TSI 3786. The modified water filled CPC which operates at a flow rate of 0.6 Litres Per Minute (LPM), is capable of detecting particles in the size range 2.5 nm to >3 μm and can measure concentrations up to $10^5$ particles $cm^{-3}$. The Cloud Condensation Nuclei (CCN) concentration was measured using a dual column Cloud Condensation Nuclei counter (CCNc; Droplet Measurement Technologies Inc. CCN-200), which is a continuous-flow stream-wise thermal gradient chamber (CFSTGC) instrument. The CCN counter operated at a flow rate of 1 LPM and the flow is evenly split between the two columns. The sample to sheath flow ratio is set to 1:10 which leaves 0.05 LPM for each column for sampling. The FAAM BAe-146 has a dedicated Rosemount inlet for providing samples to the CPC and CCNc. Before reaching the instruments the sample is passed through a Nafion dryer (Permapure MD-110-12S) operated in the atmospheric vacuum set up. The drier was installed to stop relatively warm wet ambient air from condensing in the cool conditioned cabin sample lines. It was not used to force dry the sample to some set standard. The CCNc was operated behind a constant pressure inlet set to 400 hPa. This allows a constant supersaturation to be maintained between sea level and approximately 7000 m. All the supersaturation data and concentration data are corrected for this constant pressure using Roberts et al (2010) for the former and a simple density correction for the latter.

To gain as much information as possible from a flight, the instrument is set up to have one column scanning three different supersaturations (nominally 0.79, 0.58 and 0.38% though these will change after pressure corrections) and the other stable (0.3% SS). This returns CCN concentrations for four different supersaturations every 15 mins and keeps one channel as a reference over the entire project. Internally a bleed air is removed from the sample line and trickled over a Rotronic HC2-IExxx Screw-in Relative Humidity (RH) and temperature sensor. This provides the sample RH and temperature measurements used here.

The mean ambient RH for the pre monsoon portion of the campaign is 49.7% this in itself is a fairly irrelevant measurement but is used to compare to the mean CCN sample line RH that is 28.7 %. This shows a big reduction in RH, which is due to both the nafion and the equilibrium of the ambient air to the temperature conditions of the in-cabin sample line. The effect of this and the Nafion can be untangled to some extent by using the Magnus equation to solve RH for ambient dew point and cabin influenced sample line temperature. The data can be seen in the box plot (Figure S8) and is labelled 'theoretical' which represents the theoretical values of RH in the sample line, if the nafion is removed. The mean of this value across the three pre-monsoon SWAAMI flights is 36.1%. However, a big difference can be seen in the interdecile upper range where the values are close to that of condensation, the environment the nafion dryers were used to eliminate. There are some inconsistencies in the theoretical data here (that are of little consequence) which arise from the fact that the dew point temperature from the aircraft data decomposed to 1 Hz is used in conjunction with the data from the Rotronic HC2-IExxx which reports every 5 seconds. The Rotronic data also has a long lag due to the small amount of bleed air used and the probe's response time. This approach falls apart for very dynamic environments or aircraft manoeuvres where there is rapid change in water vapour or temperature. However, the data are still indicative and the effect of the drying (though less pronounced can be seen) with mean values at 54.4, 73.5 and 57.6 % for sample, ambient and theoretical respectively (Figure S9). As would be expected the values are higher during the monsoon period than prior to it.

[Figure]

**Figure S8. Boxplots showing mean (dashed green), median (solid orange), interquartile (box) and interdecile (caps) for the RH data from the three pre-onset flights (B956, B957 and B958).**

[Figure]

**Figure S9. Boxplots showing, mean (dashed green), median (solid orange) interquartile (box) and interdecile (caps) for the RH data from the six monsoon flights (B969 to B974).**

---

## Author Comment (AC2) · 22 Apr 2021

At the outset, we thank the reviewer for the meticulous review, constructive comments and the overall appreciation of the work. We have considered each of the comments carefully and revised the manuscript. Our responses to the comments, which formed the basis for the revision, are given below along with the page and line numbers in the

revised manuscript, where the revisions are incorporated.

Reply to comments of Anonymous Reviewer #2 (RC2)

General comments:

1) A more thorough review of relevant literature in the area and on the subject should be presented. Many recent studies are not mentioned, and what the recent study offers in comparison to others is not clear.

Yes, we agree. We have added a thorough review of the related literature in the revised manuscript. The page and line numbers are mentioned along with the reply to specific comments. There have been several works with CCN in India mostly ground-based focusing on case studies or long-term measurements on seasonality. However, we focus on the vertical structure of CCN characteristics within the ABL and in the free troposphere and the changes that occur in these properties as the season changes from just prior to the onset of Indian monsoon to its active phase. This urged us to restrict our literature review to those investigating the vertical variations in CCN in the previous version of the manuscript.

2) There is a complete lack of mentioning operational supersaturation levels, which is crucial for a notion of particle activation size. Without this information all discussion falls short. Also other sampling information, such as drying of the aerosol prior to CCN and CN measurement should be mentioned.

We thank the reviewer for pointing this out. We have made the necessary changes in the revised manuscript. The page and line numbers are given along with the reply to the specific comments. We have also added a section in the supplementary material explaining these in detail.

Specific comments:

1) Introduction: As the manuscript refers to CCN and hygroscopicity, the importance of chemical composition should be also discussed and a more excessive review of the

literature in the area should mentioned. To my knowledge, there are at least two recent studies focusing on CCN in the area, also taking into account chemical composition and number size distribution. Shika et al. (2020) focus on aerosol properties also during pre-monsoon and during monsoon season and implications on cloud droplet formation. Furthermore, Arub et al. (2020) characterize chemical composition and size distributions in the area of Delhi based on air masses origin and their impact on hygroscopicity and CCN formation. Singla et al. (2017) study the role of organics in CCN activation in Western Ghats, India. Also another study part of the same experiment (CAIPEEX) by Jayachandran et al. (2020a) although mentioned in the discussion section (4.1.1) general outcomes are not mentioned in the introduction, in order to put into context the present study. Finally, Jayachandran et al. (2020b) also report airborne CCN measurements across the Indo-Gangetic Plain which also are mentioned in the discussion section (4.1.1) but not mentioned in the introduction. Overall, the introduction section needs to be enriched with other relevant studies in the area so that the current study is put into context.

Complied with in the revised manuscript (page 2, lines 48-49; page 2-3, lines 56-79; page 3, lines 87-88 and pages 3-4, lines 97-100).

2) P3, L80-86: A map with the locations of the focus areas would be helpful for the reader to get an idea of the topography and type of environment and possible aerosol sources which can impact aerosol size and chemical composition.

We have modified figure 1 in the manuscript to clearly show all the important locations. We believe that after the modification, the location of the important areas along with the direction of the wind would provide a fair idea of the environment and the possible sources.

3a) P4, L100-108: Is there a drier at the inlet?

Yes, the CCN and CPC flown during SWAAMI sampled through a Nafion dryer (Permapure MD-110-12S). This Nafion dryer was operated in the atmospheric vacuum set up.

The drier was installed to stop relatively warm, wet ambient air from condensing in the cool conditioned cabin sample lines. It was not used to force dry the sample to some set standard. Internally a bleed air is removed from the sample line and trickled over a Rotronic HC2-IExxx Screw-in Relative Humidity (RH) and temperature sensor. This provides the sample RH and temperature measurements used here. Changes made in the revised manuscript (page 5, lines 136-145).

3b) What is the RH of the sampled aerosol which enters inside the CCN counter and the CPC?

The mean RH for the pre-monsoon portion of the campaign are 49.7% and 28.7 % for the ambient and CCN sample lines respectively. This shows a big reduction in RH, which is due to both the nafion and the equilibrium of the ambient air to the temperature conditions of the in-cabin sample line. These two effects, can be untangled to some extent by using the Magnus equation to solve the RH for ambient dew point and cabin influenced sample line temperatures. The mean RH in the sample line would be 36.1%, as shown by 'Theoretical' in Figure R1 if the nafion is removed. However, the big difference can be seen in the interdecile upper range where the values are close to that of condensation, the environment the nafion dryers were used to eliminate. As expected, the RH values were much higher and the effect of the drying less pronounced during monsoon with mean values of 54.4, 73.5 and 57.6 % for sample, ambient and theoretical respectively (Figure R2). Changes made in the revised manuscript (page 5, lines 136-145).

3c) Also how was the CCN instrument operated? Was it on scanning flow analysis (Moore and Nenes, 2009; Moore et al., 2012; Lathem et al., 2013)? To my knowledge, this is the most appropriate analysis for airborne measurements as it ensures the correct supersaturation spectra over very limited timescales. If not, the CCN analysis by staying at a constant supersaturation for a given time allows for a complete CCN spectrum every, say, hour, during which obviously the aircraft has moved on to other areas, with other aerosol characteristics and sources. Even in Trembath (2013) it is not clearly

stated how the CCN instrument supersaturation varied: "Each column supersaturation was set using the proprietary dual column CCN software (DMT inc, Boulder); the set point ranged between 0.1 and 0.5 % across all flights." p.122 Or was the CCN instrument operated in a constant supersaturation? A few details on the operating mode should be included, and how the time at each supersaturation compares in terms of aircraft velocity and distance covered.

The CCN instrument used on the BAe-146 is a dual column instrument. During the SWAAMI campaign the instrument was operated behind a constant pressure inlet set to 400 hPa. This allows a constant supersaturation to be maintained between sea level and approximately 7000 m. All the supersaturation data and concentration data are corrected for this constant pressure using Roberts et al (2010) for the former and a simple density correction for the latter. To gain as much information as possible from a flight the instrument is set up to have one column scanning three different supersaturations (here 0.12%, 0.23% and 0.34%) and the other stable (0.1% SS). This returns CCN concentrations for four different supersaturations every 15 mins and keeps one channel as a reference over the entire period. More details on the operation and relative humidity variations of the sample are provided in the supplementary material. Changes made in the revised manuscript (page 5, lines 136-145).

4) P5, L136-148: Figures 3 & 4: Are all provided CCN and AFs at 0.1% supersaturation? If yes, it should be clear both in the figures and the text. There is no mention whatsoever of the instrument supersaturation in the text, not at what particle sizes this supersaturation corresponds to.

and

5) P6, L168-174: Once more, no mention of instrument supersaturation. Was it constant? Was it the same during all flights for which the ARs are compared between sites?

Figures 3 and 4 represent the CCN counts at 0.1% supersaturation. The mean supersaturation was 0.099 ±0.005 (mean ± standard deviation), with 95 % of the values within the interval 0.098 to 0.100, and with minimum and maximum values of 0.07 and 0.11 respectively. The figures and text are modified in the revised manuscript and the supersaturation is clearly mentioned (page 6, lines 171-176; page 24, Figure 3 and lines 714-716). The value of supersaturation is now included in figures 5 & 6 as well (pages 27-28). We do not have size resolved CCN measurements and the Rosemount inlet limits the maximum size of particles measures to ∼3 $\mu$m. The CCN values reported in our study represent the size-integrated CCN concentration for input particles below 3 $\mu$m, but practically the size of the particles entering the CCN counter was less than 600 nm. Passive Cavity Aerosol Spectrometer Probe (PCASP) measurements (not included in this study) showed that the optical diameters of the particles rarely crossed 600 nm and the numbers of particles above this size was negligibly small. Including the size distribution from PCASP measurements will not address the changes happening below 100 nm (particle diameter), relevant to CCN activation, hence not attempted here. The same supersaturation settings were used throughout the campaign.

6) P6, L181-184: When evoking the anthropogenic impact, anthropogenically impacted emissions are mostly in the lower particle sizes, which means that particles indeed activate in lower ranges of supersaturation. Once more the instrument supersaturation and respective particle size range should be clearly stated.

CCN counts at 0.1% supersaturation are reported here and the corresponding particle size ranges are lower than 600 nm. In general particles are activated if they have sizes larger than the critical diameter required for activation at a given supersaturation. Anthropogenic particles are smaller in size and are activated even at lower ranges of supersaturation, depending on the particle size. If they are big enough they activate at a relatively lower supersaturation. The supersaturation is mentioned in page 6, lines 171-173 and page 8, line 226.

7) P7, L203-214: All this discussion should be put in context also with particle size. Sulfate is mostly found in particle sizes larger than organics.

Complied with in the revised manuscript. The CCN activation shows strong dependence on the size of the particles. Deng et al 2011 showed that the activation curves for ambient aerosols vary significantly from those for ammonium sulfate particles. Following this study, in the range of supersaturation used in our study (0.1%) the particle diameter varies between 75 and 125 nm. If ammonium sulphate and adipic acid with size of 100 nm are considered the former can be activated at 0.15% SS compared to and 0.27% SS for the latter (Hings et al., 2008). Changes made in the revised manuscript (page 8, lines 249-251).

8) P7, L215-225: Therefore the current study offers insight of what happens above the boundary layer? This is the difference between the other studies (Brooks et al., 2019a; Jayachandran et al., 2020a)? This should be clarified, even in the introduction section.

Complied with in the revised manuscript (page 3-4, lines 97-100). The current study provides height resolved information of CCN activation up to 6 km whereas Jayachandran et al, (2020a) made measurements only up to 3.5 km. We observed the lowering of activation ratios due to the presence of BC above the boundary layer, similar to what Jayachandran et al, (2020a) observed close to the surface. Our study explores the impact of the changes in chemistry reported by Brooks et al, (2020a) on the CCN activation across the IGP. These are explicitly mentioned in the revised introduction.

9) P10, L294-195: Operational mode and settings should be comparable to those during the prior monsoon period, correct? Otherwise no comparison is possible.

and

10) Figures 8 & 9: It is clear from these figures that the CCN instrument was operated in different supersaturation levels, therefore it becomes even more imperative that the whole discussion on ARs is clarified, as well as operational conditions between pre-monsoon and monsoon flights. Also the scatter in these figures is sometimes so high, which raises confidence issues concerning the fitting (e.g. Fig. 8 d & e, 9c)

Yes, the same operational mode was followed throughout the SWAAMI campaign. The operational mode and the supersaturation has been clearly mentioned in the revised manuscript (page 5, lines 136-142). We agree that figures 8 and 9 have high scatter. Our main objective here is to show the relative changes in the properties of CCN across the IGP and not to quantify these changes in terms of their absolute magnitudes. We feel that the figures hold good in this respect. The scatter can be reduced to a certain extent by re-plotting (Figure R3) the mean CCN count against the respective supersaturation bins after omitting a few outliers. This leaves us with limited number of data points as the range of supersaturation was restricted from 0.1 to 0.4, which enabled high resolution measurements even at higher altitudes. The general observations do not change after re-plotting but the robustness of the fit is still questionable considering the limited number of points especially in the case of figure 9. The modified figure 8 is shown below (figure R3). The general pattern remains almost the same except in panels a & f having limited number of points and large spread in the data respectively. Hence, after considering the different options we decided to retain the original figures 8 and 9.

Technical corrections P7, L218: precursor gases (one word) Corrected in the revised manuscript (page 9, line 265)

We thank the reviewer for the detailed comments.

References

Deng, Z. Z., Zhao, C. S., Ma, N., Liu, P. F., Ran, L., Xu, W. Y., Chen, J., Liang, Z., Liang, S., and Huang, M. Y.: Size-resolved and bulk activation properties of aerosols in the North China Plain, Atmospheric Chemistry and Physics, 11, 3835-3846, 2011. Hings, S. S., Wrobel, W. C., Cross, E. S., Worsnop, D. R., Davidovits, P., and Onasch, T. B.: CCN activation experiments with adipic acid: effect of particle phase and adipic acid coatings on soluble and insoluble particles, Atmospheric Chemistry and Physics, 8, 3735-3748, 2008. Jayachandran, V. N., Suresh Babu, S. N., Vaishya, A., Gogoi,

M. M., Nair, V. S., Satheesh, S. K., and Krishna Moorthy, K.: Altitude profiles of cloud condensation nuclei characteristics across the Indo-Gangetic Plain prior to the onset of the Indian summer monsoon, Atmospheric Chemistry & Physics, 20, 2020. Roberts, G. C., Day, D. A., Russell, L. M., Dunlea, E. J., Jimenez, J. L., Tomlinson, J. M., Collins, D. R., Shinozuka, Y., and Clarke, A. D.: Characterization of particle cloud droplet activity and composition in the free troposphere and the boundary layer during INTEX-B, Atmospheric Chemistry and Physics, 10, 6627-6644, 2010.

Please also note the supplement to this comment:
https://acp.copernicus.org/preprints/acp-2020-1233/acp-2020-1233-AC2-supplement.pdf
* * *
[Figure]

**Fig. 1.** Figure R1. Boxplots showing, mean (dashed green), median (solid orange) interquartile (box) and interdecile (caps) for the RH data from the three pre-monsoon flights (B956, B957 and B958).

**Fig. 2.** Figure R2. Boxplots showing mean (dashed green), median (solid orange), interquartile (box) and interdecile (caps) for the RH data from the six monsoon flights (B969 to B974).

[Figure]

**Fig. 3.** Figure R3. Figure 8 in the manuscript has been re-plotted taking mean CCN versus supersaturation.